# FlyLoRA: Boosting Task Decoupling and Parameter Efficiency via Implicit Rank-Wise Mixture-of-Experts

**Heming Zou**[1][*]  **Yunliang Zang**[2][*]  **Wutong Xu**[1]  **Yao Zhu**[1]  **Xiangyang Ji**[1][†]

[1]Department of Automation, Tsinghua University
[2]Academy of Medical Engineering and Translational Medicine, Tianjin University
{zouhm24, xwt22}@mails.tsinghua.edu.cn
yunliangzang@tju.edu.cn, ee_zhuy@zju.edu.cn
xyji@tsinghua.edu.cn

## Abstract

Low-Rank Adaptation (LoRA) is a widely used parameter-efficient fine-tuning method for foundation models, but it suffers from parameter interference, resulting in suboptimal performance. Although Mixture-of-Experts (MoE)-based LoRA variants show promise in mitigating intra-task correlations in single-task instruction tuning, they introduce additional router parameters and remain ineffective in multi-task model merging where inter-task interference arises. Inspired by the fly olfactory circuit, we propose FlyLoRA, an implicit MoE-based LoRA variant that introduces: (1) rank-wise expert activation in the up-projection matrix, and (2) an implicit router that unifies expert routing and down-projection, where a frozen sparse random projection matrix replaces the traditional dense trainable version. This design resolves the trade-off between intra-task decorrelation and computational efficiency by eliminating the need for an explicit router, while inherently mitigating inter-task interference due to the orthogonality property of random matrices. Extensive experiments across four domains—general knowledge understanding, scientific question answering, mathematical reasoning, and code generation—demonstrate consistent performance improvements over existing methods. Beyond empirical gains, FlyLoRA highlights how biological structures can inspire innovations in AI technologies. Code is available at https://github.com/gfyddha/FlyLoRA.

## 1 Introduction

Foundation models have demonstrated remarkable cross-domain capabilities with the scaling of model parameters [1, 4, 15, 43, 57, 63, 64]. To enhance their performance on downstream tasks, Supervised Fine-Tuning (SFT) has become a typical post-training approach. However, full-parameter fine-tuning (Full FT) incurs prohibitive computational overhead and storage costs, making customized deployment impractical for most individual users. To address this issue, Parameter-Efficient Fine-Tuning (PEFT) [26, 27, 35, 40, 44, 47, 84, 90] has emerged as a widely adopted technique that significantly reduces resource consumption by keeping pre-trained weights frozen while fine-tuning only a small set of additional injected parameters.

Low-Rank Adaptation (LoRA) [27] is one of the most prominent PEFT methods. By leveraging the intrinsic low-dimensional properties of large language models [2, 36], LoRA approximates the parameter matrix update $\Delta W \in \mathbb{R}^{m \times n}$ as the product of two low-rank matrices, $B \in \mathbb{R}^{m \times r}$ and $A \in \mathbb{R}^{r \times n}$, where $r \ll \min(m, n)$. This method preserves much of the capability of Full FT across most tasks while substantially reducing both memory requirements and computational overhead.

---

[*]Equal contribution
[†]Corresponding author

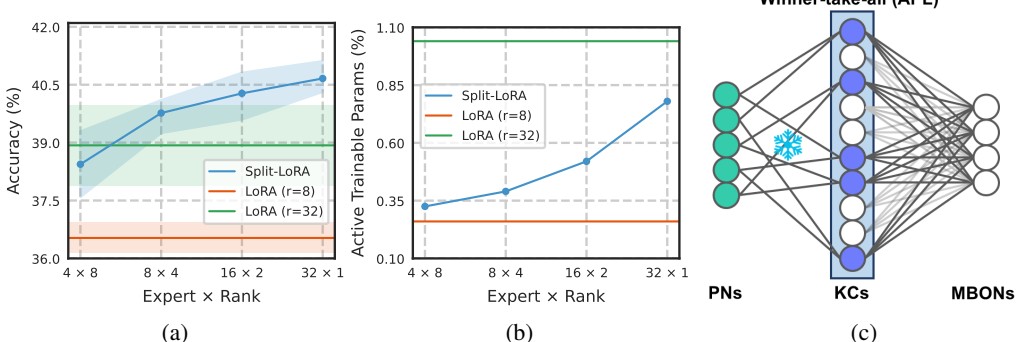

Figure 1: **(a)** Accuracy comparison under a fixed total rank $r = 32$ and activation rank $k = 8$. Finer-grained rank allocation (from 4 experts $\times$ 8 rank to 32 experts $\times$ 1 rank) yields consistent performance gains. **(b)** Activated trainable parameters (relative to Full FT) under the same budget. Increasing expert granularity leads to a monotonic rise in activated parameters due to router overhead. **(c)** Schematic of the fly olfactory circuit. Odor signals in projection neurons (PNs) are randomly projected to Kenyon cells (KCs), with each KC connecting to a fixed number of PNs (but not all), forming sparse connections. These signals are then selectively projected to mushroom body output neurons (MBONs), while lateral inhibition from an anterior paired lateral (APL) neuron suppresses weak KC-MBON connections, implementing a winner-take-all strategy. Thus, the number of activated KCs is much smaller than the total dimension of the KC layer.

However, to achieve strong performance on complex tasks, LoRA typically requires much higher ranks, which contradicts PEFT's core goal of efficiency [31, 45]. Moreover, interference within LoRA's ranks can impair training [76], leading to issues such as hallucination [22] and gradient explosion [58], thereby largely limiting its potential. We refer to this challenge as **intra-task interference**. Meanwhile, foundation models often need to integrate multiple capabilities to handle complex downstream tasks, but retraining on multi-domain corpora is expensive [71], particularly when several specialized models are already available. Consequently, model merging [11, 28, 29, 54] is widely used to combine LoRA components trained on different domains in a training-free manner. Arising from conflicts between different components, this process introduces another challenge: **inter-task interference**.

To address intra-task interference, several studies have incorporated the Mixture-of-Experts (MoE) architecture [19, 30, 34, 70, 78, 81] into LoRA [17, 20, 21, 37, 55, 76, 82, 89], where each expert learns specialized knowledge to partially achieve task decoupling. We refer to these approaches as **MoE-based LoRA methods**. They replace the original low-rank matrices with multiple experts and use a dynamic router to selectively activate them. By leveraging redundant parameters and sparse activations, these methods keep the computational budget comparable to LoRA with fewer total ranks. However, they may still suffer from interference within each expert. To investigate this issue, we conduct pilot studies on MMLU using Llama-3.1-8B, adopting Split-LoRA as a representative MoE-based method. As shown in Figure 1(a), finer-grained rank allocation yields consistent performance improvements under a fixed budget. However, as illustrated in Figure 1(b), pushing expert granularity to extremes also increases the number of activated trainable parameters due to additional router overhead. This trade-off makes it difficult to achieve both high performance and efficiency. Meanwhile, resolving inter-task interference has received limited emphasis in existing MoE-based LoRA methods.

Therefore, we seek to design an improved MoE-based LoRA variant that simultaneously achieves:

- *Reduced parameter interference among different ranks within a single LoRA component;*

- *Reduced parameter interference between different LoRA components;*

- *Reduced activated trainable parameters in routers.*

Inspired by the fly olfactory circuit [6, 13, 38, 42, 72, 100], which shows strong similarity to MoE-based LoRA, we introduce an implicit router to mitigate the trade-off between intra-task interference and efficiency. As shown in Figure 1(c), this leads to the design of **FlyLoRA**, which (1) treats matrix $\boldsymbol{A}$ as a frozen sparse random projection that maps inputs into a higher-rank space (e.g., $r = 32$ vs.

$r = 8$); and (2) simulates the bio-inspired "winner-take-all" mechanism by activating $k$ rank-1 experts in $\boldsymbol{B}$ linked to the top-$k$ magnitudes after projection by $\boldsymbol{A}$. This unifies the roles of $\boldsymbol{A}$ and router $\boldsymbol{G}$ into a single frozen projection, jointly performing down-projection and expert selection. Without explicit router parameters, the resulting implicit rank-wise MoE structure maintains computational efficiency while reducing intra-task interference. Moreover, we theoretically show that distinct random projections $\boldsymbol{A}_i$ and $\boldsymbol{A}_j$ from different LoRA components naturally map task updates into approximately orthogonal subspaces, thereby alleviating inter-task interference.

In summary, the core contributions of our proposed FlyLoRA framework are:

• **Efficient Intra-task Decoupling:** By using implicit rank-wise MoE, we enable finer expert allocation with reduced parameter interference in single-task scenarios. Additionally, FlyLoRA surpasses MoE-based LoRA in efficiency by eliminating the need for router parameters.

• **Efficient Inter-task Decoupling:** In multi-task model merging scenarios, different random projections $\boldsymbol{A}_i$ and $\boldsymbol{A}_j$ naturally form approximately orthogonal subspaces. This inherent property ensures different LoRA components operate in uncorrelated subspaces, thus achieves decoupling.

• **Neuroscience-Inspired Design:** The efficacy of our algorithm, combined with its structural alignment with the fly olfactory circuit, establishes a promising bridge between neuroscience and artificial intelligence.

## 2 Revisiting MoE-based LoRA Methods

### 2.1 Preliminaries

LoRA (visualized in Figure 2(a)) simulates weight updates during fine-tuning by decomposing the update matrix into two learnable low-rank matrices. Given a pretrained weight matrix $\boldsymbol{W}_0 \in \mathbb{R}^{m \times n}$, the parameter update is computed as:

$$\boldsymbol{W}' = \boldsymbol{W}_0 + \Delta \boldsymbol{W} = \boldsymbol{W}_0 + \frac{\alpha}{r} \boldsymbol{B} \boldsymbol{A}, \tag{1}$$

where $\boldsymbol{B} \in \mathbb{R}^{m \times r}$, $\boldsymbol{A} \in \mathbb{R}^{r \times n}$, and the rank $r \ll \min(m, n)$. The scaling factor $\alpha$ is typically set to $2r$. For an input embedding $\boldsymbol{x} \in \mathbb{R}^n$, the forward pass becomes:

$$f_{\text{LoRA}}(\boldsymbol{x}) = \boldsymbol{W}' \boldsymbol{x} = \boldsymbol{W}_0 \boldsymbol{x} + \frac{\alpha}{r} \boldsymbol{B} \boldsymbol{A} \boldsymbol{x}. \tag{2}$$

Here, $\boldsymbol{W}_0$ remains frozen during training, while only $\{\boldsymbol{A}, \boldsymbol{B}\}$ are updated. This approach reduces the number of trainable parameters from $\mathcal{O}(mn)$ to $\mathcal{O}(r(m + n))$, thereby achieving higher parameter efficiency. The low-rank structure allows LoRA to maintain stable performance while significantly reducing both computational overhead and GPU memory requirements during fine-tuning.

### 2.2 MoE-based LoRA Framework

The MoE paradigm (visualized in Figure 2(b)) extends LoRA by decomposing the low-rank adaptation into $N$ specialized experts. Each expert $\boldsymbol{E}_i$ is parameterized by a pair of matrices $\{\boldsymbol{B}_i \in \mathbb{R}^{m \times r_i}, \boldsymbol{A}_i \in \mathbb{R}^{r_i \times n}\}$, where $r_i$ denotes the expert-specific rank. The forward pass incorporates a gating mechanism $\boldsymbol{G}(\boldsymbol{x}) : \mathbb{R}^n \to \mathbb{R}^N$ that dynamically routes inputs to activate the most relevant experts. Formally, the output combines the frozen pretrained weights $\boldsymbol{W}_0$ with a sparse combination of expert contributions:

$$f_{\text{MoE-LoRA}}(\boldsymbol{x}) = \boldsymbol{W}_0 \boldsymbol{x} + \frac{\alpha}{r} \sum_{i=1}^{N} \boldsymbol{G}(\boldsymbol{x})_i \cdot \underbrace{\boldsymbol{B}_i \boldsymbol{A}_i \boldsymbol{x}}_{\boldsymbol{E}_i(\boldsymbol{x})}, \tag{3}$$

where the router $\boldsymbol{G}(\boldsymbol{x})$ typically follows a top-$k$ selection policy via a trainable projection $\boldsymbol{W}_g \in \mathbb{R}^{N \times n}$. For simplicity, we omit the activation function, formulating the router as:

$$\boldsymbol{G}(\boldsymbol{x}) = \text{top-}k(\boldsymbol{W}_g \boldsymbol{x}). \tag{4}$$

By activating only $k$ experts per input, this design maintains computational efficiency. The sparse routing strategy enables conditional computation, which expands the model's representational capacity without incurring a proportional increase in computational cost. In our work, we implement Split-LoRA under this framework as a minimal yet representative instantiation of MoE-based LoRA. Further implementation details are provided in Appendix C.3.

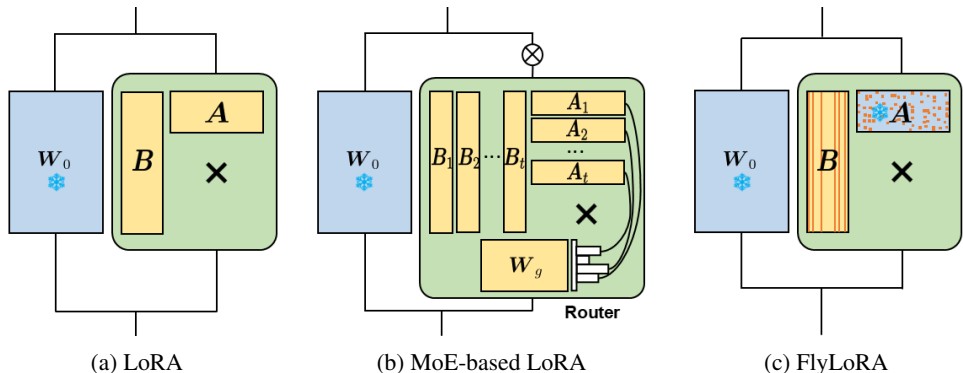

Figure 2: **Schematic illustrations of different LoRA variants.** **(a)** LoRA employs low-rank matrices $A$ and $B$ to simulate weight updates, where each row of $A$ is fully connected to the corresponding column of $B$. **(b)** MoE-based LoRA decomposes the updates into multiple small experts $\{A_i, B_i\}_{i=1}^{N}$ and uses a router to determine which experts should be activated. **(c)** FlyLoRA unifies the down-projection and router into a frozen matrix $A$ and selectively activates only the ranks in $B$ linked to the top-$k$ magnitude activations after projection through $A$.

### 2.3 Pushing MoE-based LoRA Architecture to the Extreme

Comparing Eq. 2 and Eq. 3 reveals that MoE-based LoRA can be viewed as a finer-grained, sparsely activated variant of LoRA, where the separation of experts mitigates task conflicts. Taking this decomposition to the extreme motivates our **rank-wise expert** design, where each expert governs a single rank, achieving the best decorrelating effect (see Figure 1(a)). Formally, for a rank-$r$ LoRA, the matrices $A$ and $B$ can be decomposed into $r$ rank-1 components:

$$f_{\text{rank-wise-LoRA}}(x) = W_0 x + \frac{\alpha}{r} \sum_{i=1}^{r} G(x)_i \cdot \underbrace{b_i a_i x}_{E_i(x)}, \tag{5}$$

with $a_i = A[i,:] \in \mathbb{R}^{1 \times n}$ and $b_i = B[:,i] \in \mathbb{R}^{m \times 1}$.

However, this approach introduces a scalability challenge: the router's linear layer $W_g \in \mathbb{R}^{N \times n}$ grows linearly with the number of experts $N$ (see Figure 1(b)). Under a fixed total rank, finer-grained experts with larger $N$ make the explicit routing mechanism computationally prohibitive, undermining the efficiency gains of sparse activation.

To overcome this limitation, we seek an **implicit routing mechanism** that eliminates the need for the explicit router parameter $W_g$ entirely. This entails finding a proxy that leverages intrinsic signals within the model to select the top-$k$ experts, effectively approximating the function of the original router $G$. To address this, we draw inspiration from the perspective of Singular Value Decomposition (SVD), which can also be viewed as a rank-wise decomposition. In SVD, the low-rank update matrix $\Delta W$ can be decomposed as $\Delta W = \sum_{i=1}^{r} \sigma_i u_i v_i^\top$, where $\sigma_i$ denotes the $i$-th singular value (indicating the importance of the corresponding component), $u_i$ is the $i$-th left-singular vector, and $v_i$ is the $i$-th right-singular vector. Each component $\sigma_i u_i v_i^\top$ is a rank-1 update. The Eckart-Young-Mirsky theorem [18] guarantees that the top-$k$ components, selected based on the magnitude of $\sigma_i$, provide the best rank-$k$ approximation to the original rank-$r$ matrix in terms of Frobenius norm, thereby capturing the most salient features with minimal reconstruction error. While exact SVD is computationally prohibitive and thus impractical in our framework, this insight naturally suggests that the magnitude of each rank-1 term, $\|b_i a_i x\|$ in Eq. 6, approximately reflects its importance:

$$f_{\text{LoRA}}(x) = W_0 x + \frac{\alpha}{r} \sum_{i=1}^{r} b_i a_i x. \tag{6}$$

Nevertheless, a naive approach of first computing all $r$ terms $b_i a_i x$ and then selecting the top-$k$ would also forfeit the computational benefits of sparse activation, as the cost of computing all terms remains $\mathcal{O}(rmn)$. This necessitates a routing strategy that can identify the most important experts *before* fully computing their outputs. Furthermore, beyond efficient routing, another critical limitation of existing MoE-based LoRA methods is their lack of inherent support for multi-task deployment.

When merging models already fine-tuned on different tasks, interference between LoRA adapters often leads to significant performance degradation, as the underlying architecture does not structurally encourage task-specific updates to reside in orthogonal or non-overlapping parameter subspaces.

These dual challenges motivate the following two key design requirements for an improved MoE-based LoRA framework:

   • *Implicit magnitude-based router for top-$k$ activation, without explicit router parameters, enabling expert selection prior to full computation;*

   • *Native support for training-free model merging through architectural properties that promote inter-task interference mitigation.*

## 3  FlyLoRA

Inspired by the fly olfactory circuit (Figure 1(c)), whose neural architecture inherently meets our requirements for MoE-based LoRA variants, we propose FlyLoRA (visualized in Figure 2(c)). Section 3.1 presents its formal design, while subsequent sections analyze its key advantages: Section 3.2 shows how a fixed $A$ acts as an implicit router, Section 3.3 demonstrates intra-task decoupling, and Section 3.4 establishes inherent support for inter-task decoupling in model merging.

### 3.1  Formulation of FlyLoRA

In FlyLoRA, the matrix $A \in \mathbb{R}^{r \times n}$ is sparse and frozen. It is randomly initialized at the beginning and remains frozen during training, implementing an intrinsic top-$k$ operation in the projection space $\mathbb{R}^r$ for implicit routing. Given an input token $x \in \mathbb{R}^n$, this process is formulated as:

$$y' = \text{top-}k(y) = \text{top-}k\left(Ax\right), \tag{7}$$

where each row of $A$ contains exactly $p$ ($p < n$) non-zero entries independently sampled from $\mathcal{N}(0, \frac{1}{r^2})$ (a widely used standard initialization). We define the sparsity ratio as $\rho = \frac{p}{n}$. After projection through $A$, only the columns $b_i \in \mathbb{R}^m$ ($i \in \{1, \ldots, r\}$) in the up-projection matrix $B \in \mathbb{R}^{m \times r}$ linked to dimensions with top-$k$ ($k < r$) magnitudes in $Ax \in \mathbb{R}^r$ are activated. Formally:

$$[By']_i = \begin{cases} [By]_i & \text{if the magnitude of } [y]_i \text{ is among the top-}k \text{ values of } y, \\ 0 & \text{otherwise.} \end{cases} \tag{8}$$

To enhance training stability in this MoE structure, we incorporate a simple expert-wise bias term $d \in \mathbb{R}^r$ for loss-free load balancing, following [43]. This auxiliary term is updated manually via:

$$d_i \leftarrow d_i + u \cdot \text{sign}(\bar{c}_i - c_i), \tag{9}$$

where $u$ is a small learning rate, $\bar{c}_i$ represents the expected assignment frequency for expert $i$, $c_i$ tracks the actual assignment count, and $\text{sign}(\cdot)$ denotes the sign function. This bias term $d$ is added to $Ax$ in expert selection to promote the activation of under-activated experts and suppress over-activated experts, thereby achieving load balancing. Thus, the activated experts are selected by:

$$\mathcal{I}_{\text{top}k} = \{i_1, \ldots, i_k\} \quad \text{where} \quad i_j = \underset{i \notin \{i_1, \ldots, i_{j-1}\}}{\arg\max}\left(Ax + d\right)_i. \tag{10}$$

The forward pass is then computed as:

$$f_{\text{FlyLoRA}}(x) = W_0 x + \Delta W x = W_0 x + \frac{\alpha}{r}\sum_{i=1}^{r} \mathbb{I}(i \in \mathcal{I}_{\text{top}k}) \cdot b_i a_i x, \tag{11}$$

where $\mathbb{I}(\cdot)$ denotes the indicator function.

### 3.2  Fixed Sparse Random Projection as Implicit Router

The core objective of the MoE router is to select $k$ out of $r$ experts that best approximate the effect of using all experts. However, as established in Sec. 2.3, it is computationally impractical to determine the selection based on $\|b_i a_i x\|$. We therefore seek to perform the selection using only $a_i x \in \mathbb{R}$ as a surrogate. In FlyLoRA, since we fix $A$ as a sparse random projection, computing $a_i x$ is as efficient as in standard $\text{LoRA}_{(r=k)}$. We theoretically prove that this projection preserves pairwise distances (see Theorem 3.1), and thus can serve as an effective implicit router.

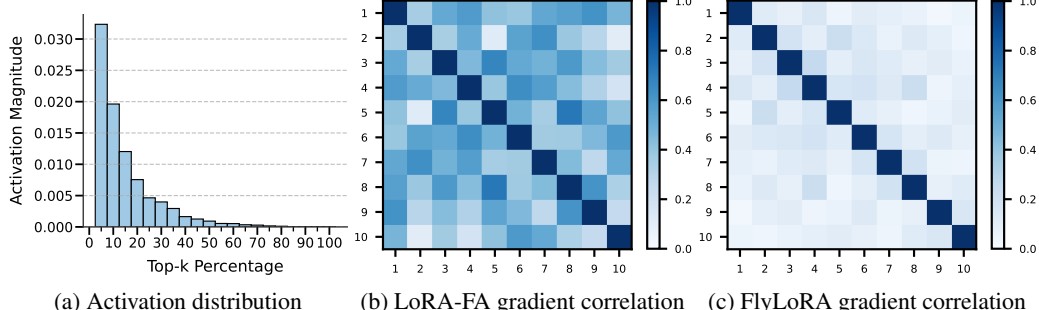

| (a) Activation distribution | (b) LoRA-FA gradient correlation | (c) FlyLoRA gradient correlation |

Figure 3: **(a)** Activation value magnitude distribution across dimensions, showing the mean activation strength at different top-$k$ selection percentages. **(b-c)** Gradient correlation matrices of (b) LoRA-FA$_{(r=32)}$ versus (c) FlyLoRA$_{(k=8)}$'s $\boldsymbol{B}$ matrices (10 randomly sampled columns). For a simplified illustration, we use the LoRA module of q_proj in the middle layer of Llama-3.1-8B on MMLU.

**Theorem 3.1.** *Given the matrix $\boldsymbol{A} \in \mathbb{R}^{r \times n}$ with each row having exactly $p$ non-zero entries randomly sampled from $\mathcal{N}(0, \frac{1}{r^2})$, for any $\epsilon > 0$,*

$$\mathbb{P}\left((1-\epsilon)\|\boldsymbol{x}-\boldsymbol{y}\|^2 \leq \frac{1}{r\sigma^2}\|\boldsymbol{A}\boldsymbol{x}-\boldsymbol{A}\boldsymbol{y}\|^2 \leq (1+\epsilon)\|\boldsymbol{x}-\boldsymbol{y}\|^2\right) \geq 1 - e^{-(\epsilon^2-\epsilon^3)\frac{r}{4}} - e^{-\frac{(\epsilon^2-\epsilon^3)r}{2(\frac{3p}{n}+1)}},$$

*for any input embeddings $\boldsymbol{x}, \boldsymbol{y} \in \mathbb{R}^n$, where $\sigma^2 = \frac{p}{nr^2}$. A detailed proof is provided in Appendix A.1.*

This theorem establishes a probabilistic guarantee for the approximate preservation of Euclidean distances under a sparse random projection. Notably, the concentration bound tightens with reduced sparsity ($p/n$) and larger rank ($r$), which aligns with intuitive expectation. We further present its robustness empirically via hyperparameter sensitivity analysis in Section 4.4.

Building on this property, we posit that FlyLoRA can self-select experts based on the values of $\boldsymbol{a}_i\boldsymbol{x}$, which aligns with the finding in [49] that "an expert is aware of its own capacity to effectively process a token, an awareness reflected in the scale of its internal activations." Unlike traditional trainable MoE routers that explicitly learn routing weights, FlyLoRA leverages the fixed geometry of $\boldsymbol{A}$ to perform implicit, activation-driven routing. This design not only eliminates the difficulty of learning routing parameters and avoids the separation between the router's decision-making and the experts' execution, but also reduces training instability by removing the stochasticity in router optimization. Because the projection $\boldsymbol{A}$ preserves pairwise distances, two semantically similar inputs $\boldsymbol{x}_i$ and $\boldsymbol{x}_j$ are mapped to nearby low-dimensional representations $\boldsymbol{A}\boldsymbol{x}_i$ and $\boldsymbol{A}\boldsymbol{x}_j$, and therefore routed to similar experts, while dissimilar inputs are routed to different experts. This geometry-induced consistency helps mitigate expert representation homogenization in MoE models, enabling each expert to focus on specialized knowledge, reduce internal conflicts, and improve sample efficiency (as supported by previous studies, e.g., [10]). In this sense, FlyLoRA resembles a variant of the hash router [66], which also achieves lightweight and stable expert assignment through a fixed mapping. Consequently, the top-$k$ operation naturally selects the most important experts according to the magnitudes of $\boldsymbol{a}_i\boldsymbol{x}$. In Figure 3(a), empirically around top-25% of dimensions account for more than 80% "energy". Thus, it typically does not cause a large performance drop.

## 3.3 Gradient Decoupling via Top-k Sparsity

In the FlyLoRA framework, only matrix $\boldsymbol{B}$ requires updating. We theoretically demonstrate that our rank-wise expert allocation strategy, induced by top-$k$ selection, inherently reduces gradient covariance between distinct experts, thus mitigating intra-task interference. Our analysis begins with Assumption 3.2 describing the sparsity pattern of activations. For analytical convenience, we consider a simplified condition where the top-$k$ operation randomly selects $k$ out of $r$ columns for activation.

**Assumption 3.2** (Uniform Sparse Activation). *During the top-$k$ operation, each training sample activates exactly $k$ columns of the parameter matrix $\boldsymbol{B} \in \mathbb{R}^{m \times r}$, with uniform selection probability $p = \frac{k}{r}$ per column.*

Based on Assumption 3.2, we derive Theorem 3.3 (proof in Appendix A.2).

**Theorem 3.3** (Covariance Reduction Under top-$k$). *Let $\tilde{\boldsymbol{\Sigma}}$ and $\boldsymbol{\Sigma}$ denote the gradient covariance matrices with and without top-$k$ activation. When $r > k$, off-diagonal entries scale as:*

$$\mathbb{E}[\tilde{\boldsymbol{\Sigma}}_{(i,j)}] \approx \mathbb{E}[\boldsymbol{\Sigma}_{(i,j)}] \cdot \frac{k^2}{r^2}, \quad \forall i \neq j.$$

This $\mathcal{O}(k^2/r^2)$ reduction factor quantifies how top-$k$ sparsity promotes parameter decoupling by suppressing interference terms. When $k = 1$ (only one rank is activated), the off-diagonal covariance almost vanishes, achieving full decoupling; when $k = r$ (all ranks are activated, degenerating to LoRA-FA [94]), it recovers the dense training regime. Theorem 3.3 is proved in Appendix A.2. To empirically validate this theoretical result, we visualize the gradient correlation patterns of LoRA-FA and FlyLoRA in Figure 3(b) and (c), where correlations are computed using 10 randomly selected gradient columns (see heatmap visualization). The observed sparsity pattern strongly supports our theoretical prediction of reduced off-diagonal covariance under top-$k$ selection.

### 3.4 Inter-Task Orthogonality in Model Merging

Traditional LoRA model merging often suffers from parameter interference when combining task-specific components through weight averaging:

$$\boldsymbol{W}' = \boldsymbol{W}_0 + \sum_{i=1}^{t} w_i \boldsymbol{B}_i \boldsymbol{A}_i. \tag{12}$$

We analyze how FlyLoRA's inherent subspace orthogonality enables effective multi-task model merging. We derive Theorem 3.4 (proof in Appendix A.3).

**Theorem 3.4** (Approximate Subspace Orthogonality). *For independent random matrices $\boldsymbol{A}_i, \boldsymbol{A}_j \in \mathbb{R}^{r \times n}$ with sparse Gaussian entries ($\mathcal{N}(0, \frac{1}{r^2})$ for $p < n$ randomly selected entries per row), the following holds,*

1. ***Exact mean orthogonality**: $\mathbb{E}[\boldsymbol{A}_i \boldsymbol{A}_j^\top] = \boldsymbol{0}_{r \times r}$*

2. ***Polynomially decaying correlations**: $\mathbb{P}(\|\boldsymbol{A}_i \boldsymbol{A}_j^\top\|_2 \geq \epsilon r) \leq \frac{p^2}{nr^2\epsilon^2}$*

This theorem establishes that sparse random projections naturally induce nearly orthogonal subspaces. The residual correlation bound of order $\mathcal{O}(\frac{p^2}{nr^2})$ indicates that interference becomes negligible under practical parameter scales. This property directly leads to Corollary 3.5 (proof in Appendix A.3).

**Corollary 3.5.** *Let $\boldsymbol{A}_i, \boldsymbol{A}_j \in \mathbb{R}^{r \times n}$ be fixed sparse random projections after initialization. Then for any learned matrices $\boldsymbol{B}_i \boldsymbol{A}_i$ and $\boldsymbol{B}_j \boldsymbol{A}_j$, they satisfy the pairwise orthogonality property:*

$$\langle \boldsymbol{B}_i \boldsymbol{A}_i, \boldsymbol{B}_j \boldsymbol{A}_j \rangle_F \approx 0 \quad for \quad i \neq j.$$

This orthogonal decomposition provides a key advantage for model merging: task-specific updates $\boldsymbol{B}_i \boldsymbol{A}_i$ occupy nearly orthogonal subspaces, thereby preventing destructive interference. The "Pairwise Orthogonality" property captures FlyLoRA's behavior during multi-task aggregation. According to geometric intuition that orthogonality facilitates model merging [29, 56, 75], and consistent with theoretical analyses in [39, 92], FlyLoRA's fixed sparse projection design aligns with this principle. Empirically (see Section 4.3), the random projection $\boldsymbol{A}$ enables FlyLoRA to preserve task-specific performance after merging, whereas the learnable $\boldsymbol{A}$ in conventional LoRA exhibits significantly higher interference. A similar analysis of LoRA merging was conducted in a concurrent study [93].

## 4 Experiments

### 4.1 Experimental Setup

**Datasets and Backbones:** We evaluate FlyLoRA's performance across four key domains: (1) *general knowledge understanding* using the MMLU [25] benchmark with auxiliary training datasets for fine-tuning and test set for evaluation, (2) *scientific question answering* using the ScienceQA [48] dataset for fine-tuning and evaluation, (3) *mathematical reasoning* on GSM8K [12] problems for

Table 1: **Performance Comparison of LoRA Variants in Single-task Evaluation.** We evaluate various methods across four benchmarks: MMLU, ScienceQA, GSM8K (accuracy), and HumanEval (Pass@k), with all metrics reported in percentage (%). Param(%) indicates the percentage of activated trainable parameters relative to Full FT. The best results are highlighted in **bold**.

| Model | Method | Param(%) | MMLU | ScienceQA | GSM8K | HumanEval | | |
|---|---|---|---|---|---|---|---|---|
| | | | | | | Pass@1 | Pass@5 | Pass@10 |
| Llama-3.1-8B | LoRA$_{(r=8)}$ | 0.26 | 36.53$_{\pm0.40}$ | 91.39$_{\pm0.55}$ | 55.34$_{\pm0.24}$ | 29.13$_{\pm0.56}$ | 52.28$_{\pm1.24}$ | 61.67$_{\pm0.61}$ |
| | LoRA$_{(r=32)}$ | 1.03 | 38.93$_{\pm1.04}$ | 94.01$_{\pm0.17}$ | 56.25$_{\pm0.29}$ | 30.37$_{\pm1.06}$ | 54.37$_{\pm0.39}$ | 64.02$_{\pm0.94}$ |
| | Split-LoRA$_{(4\times8)}$ | 0.33 | 38.44$_{\pm0.69}$ | 92.41$_{\pm0.54}$ | 55.65$_{\pm0.47}$ | 31.28$_{\pm1.52}$ | 54.16$_{\pm1.12}$ | 63.94$_{\pm0.89}$ |
| | **FlyLoRA$_{(k=8)}$** | **0.13** | **40.88$_{\pm1.61}$** | **94.15$_{\pm0.36}$** | **58.76$_{\pm0.74}$** | **36.88$_{\pm1.91}$** | **62.40$_{\pm1.82}$** | **73.34$_{\pm1.24}$** |
| Qwen-2.5-7B | LoRA$_{(r=8)}$ | 0.26 | 49.84$_{\pm0.56}$ | 92.84$_{\pm0.13}$ | 77.01$_{\pm0.32}$ | 47.20$_{\pm1.54}$ | 78.89$_{\pm0.36}$ | 85.94$_{\pm0.64}$ |
| | LoRA$_{(r=32)}$ | 1.05 | 52.07$_{\pm0.31}$ | 95.01$_{\pm0.21}$ | 79.23$_{\pm0.22}$ | 52.87$_{\pm1.79}$ | 81.67$_{\pm1.14}$ | 87.80$_{\pm0.72}$ |
| | Split-LoRA$_{(4\times8)}$ | 0.33 | 50.68$_{\pm1.06}$ | 93.08$_{\pm0.41}$ | 77.12$_{\pm0.76}$ | 48.65$_{\pm1.18}$ | 79.30$_{\pm0.91}$ | 86.05$_{\pm0.44}$ |
| | **FlyLoRA$_{(k=8)}$** | **0.13** | **53.68$_{\pm0.47}$** | **95.55$_{\pm0.18}$** | **80.82$_{\pm0.56}$** | **54.34$_{\pm2.13}$** | **82.85$_{\pm0.52}$** | **89.63$_{\pm0.55}$** |

fine-tuning and evaluation, and (4) *code generation* assessed via CodeAlpaca-20k [7] for training and HumanEval [9] for evaluation. Except for HumanEval, which is evaluated via pass@k metrics, others are evaluated via accuracy. All benchmarks are evaluated in a zero-shot manner. We examine our framework in both single-task configurations, training with these four datasets individually, and multi-task settings, where the LoRA components trained in single-task setup for each dataset are merged together in a training-free manner. Most experiments are conducted using Llama-3.1-8B [23] and Qwen-2.5-7B [86], respectively. See Appendix C for implementation details.

**Baselines:** For the single-task setup, we compare FlyLoRA against: (1) vanilla LoRA with identical activation ranks (LoRA$_{(r=8)}$) and total ranks (LoRA$_{(r=32)}$), and (2) representative MoE-based LoRA variants Split-LoRA$_{(4\times8)}$ (abbreviated for 4 expert$\times$8 rank). For the multi-task setup, we benchmark FlyLoRA against them using weight averaging fusion, and several advanced merging techniques. Across all datasets, FlyLoRA$_{(k=8)}$ uses total rank $r = 32$ but activates only $k = 8$ ranks after the top-$k$ operation between $A$ and $B$, with the fixed sparse random $A$'s sparsity ratio $\rho$ set to $8/32$.

### 4.2 Single Task Performance

The single-task results are presented in Table 1. Despite operating under a lower computational budget, FlyLoRA$_{(k=8)}$ outperforms LoRA variants with the same rank (LoRA$_{(r=8)}$) across all datasets. This improvement can be attributed to its broader parameter space. Notably, FlyLoRA$_{(k=8)}$ also achieves slightly better performance than LoRA variants with the same total rank (LoRA$_{(r=32)}$), suggesting that a significant portion of LoRA's parameters are redundant and may introduce interference. Additionally, FlyLoRA$_{(k=8)}$ demonstrates superior performance over Split-LoRA$_{(4\times8)}$, highlighting the benefits of its finer expert allocation strategy within the MoE framework, which enables **intra-task decoupling**. The reduction in activated trainable parameters compared to these baselines shows FlyLoRA's efficiency. Extended results with larger models and further baselines are in Appendix B.

### 4.3 Multi-task Performance

For simplicity, we first employ the widely used weight averaging technique for model merging. Specifically, this corresponds to setting $w_i = \frac{1}{t}$ in Eq. 12, yielding the merged weights $W' = W_0 + \frac{1}{t}\sum_{i=1}^{t} B_i A_i$. The multi-task results are presented in Table 2, where LoRA components from different domains are merged. Compared to both LoRA variants ($r = 8$ and $r = 32$) and Split-LoRA$_{(4\times8)}$, FlyLoRA achieves higher accuracy both before and after merging, with significantly smaller performance degradation. This robustness stems from its **inter-task decoupling** enabled by approximate orthogonal random projection, as theoretically analyzed in Section 3.4. Additional results with advanced fusion techniques are provided in Appendix B.

### 4.4 Ablation Study and Hyperparameter Sensitivity Analysis

We conduct an ablation study to analyze key properties of FlyLoRA by evaluating two critical modifications: (1) removing load-balancing strategies and (2) replacing the frozen matrix $A$ with an

Table 2: **Multi-task Performance Comparison Before and After Parameter Merging.** We evaluate LoRA variants across MMLU, ScienceQA, GSM8K (accuracy), and HumanEval (Pass@k) benchmarks. The table shows performance before merging, after merging, and the relative performance drop ($\Delta\%$). The best results are highlighted in **bold**.

| Model | Method | Merge Status | MMLU | ScienceQA | GSM8K | HumanEval Pass@1 | Pass@5 | Pass@10 |
|---|---|---|---|---|---|---|---|---|
| Llama-3.1-8B | $\text{LoRA}_{(r=8)}$ | Before | $36.53_{\pm0.40}$ | $91.39_{\pm0.55}$ | $55.34_{\pm0.24}$ | $29.13_{\pm0.56}$ | $52.28_{\pm1.24}$ | $61.67_{\pm0.61}$ |
| | | After | $30.05_{\pm0.82}$ | $31.05_{\pm2.38}$ | $25.19_{\pm2.36}$ | $16.09_{\pm3.15}$ | $45.38_{\pm1.62}$ | $56.49_{\pm2.13}$ |
| | | $\Delta\,(\%)$ | -6.48 | -60.34 | -30.15 | -13.04 | -6.90 | -5.18 |
| | $\text{LoRA}_{(r=32)}$ | Before | $38.93_{\pm1.04}$ | $94.01_{\pm0.17}$ | $56.25_{\pm0.29}$ | $30.37_{\pm1.06}$ | $54.37_{\pm0.39}$ | $64.02_{\pm0.94}$ |
| | | After | $34.02_{\pm1.32}$ | $34.35_{\pm1.42}$ | $24.77_{\pm0.94}$ | $18.94_{\pm1.48}$ | $46.39_{\pm1.74}$ | $59.27_{\pm1.19}$ |
| | | $\Delta\,(\%)$ | -4.91 | -59.66 | -31.48 | -11.43 | -7.98 | -4.75 |
| | $\text{Split-LoRA}_{(4\times8)}$ | Before | $38.44_{\pm0.69}$ | $92.41_{\pm0.54}$ | $55.65_{\pm0.47}$ | $31.28_{\pm1.52}$ | $54.16_{\pm1.12}$ | $63.94_{\pm0.89}$ |
| | | After | $33.58_{\pm1.16}$ | $37.67_{\pm1.06}$ | $27.35_{\pm1.10}$ | $21.36_{\pm1.08}$ | $46.01_{\pm0.87}$ | $59.52_{\pm0.95}$ |
| | | $\Delta\,(\%)$ | -4.86 | -54.74 | -28.30 | -9.92 | -8.15 | -4.42 |
| | $\text{FlyLoRA}_{(k=8)}$ | Before | $40.88_{\pm1.61}$ | $94.15_{\pm0.36}$ | $58.76_{\pm0.74}$ | $36.88_{\pm1.91}$ | $62.40_{\pm1.82}$ | $73.34_{\pm1.24}$ |
| | | After | $38.86_{\pm1.46}$ | $51.10_{\pm0.71}$ | $36.95_{\pm1.37}$ | $32.61_{\pm1.45}$ | $56.59_{\pm2.66}$ | $69.76_{\pm0.75}$ |
| | | $\Delta\,(\%)$ | **-2.02** | **-43.05** | **-21.81** | **-4.27** | **-5.81** | **-3.58** |
| Qwen-2.5-7B | $\text{LoRA}_{(r=8)}$ | Before | $49.84_{\pm0.56}$ | $92.84_{\pm0.13}$ | $77.01_{\pm0.32}$ | $47.20_{\pm1.54}$ | $78.89_{\pm0.36}$ | $85.94_{\pm0.64}$ |
| | | After | $44.62_{\pm1.23}$ | $60.07_{\pm2.18}$ | $81.56_{\pm1.48}$ | $22.09_{\pm2.68}$ | $68.38_{\pm0.86}$ | $80.49_{\pm0.77}$ |
| | | $\Delta\,(\%)$ | -5.22 | -32.77 | +4.55 | -25.21 | -10.51 | -5.45 |
| | $\text{LoRA}_{(r=32)}$ | Before | $52.07_{\pm0.31}$ | $95.01_{\pm0.21}$ | $79.23_{\pm0.22}$ | $52.87_{\pm1.79}$ | $81.67_{\pm1.14}$ | $87.80_{\pm0.72}$ |
| | | After | $32.86_{\pm1.06}$ | $55.58_{\pm0.76}$ | $83.91_{\pm0.70}$ | $23.84_{\pm2.13}$ | $66.23_{\pm1.65}$ | $79.27_{\pm1.05}$ |
| | | $\Delta\,(\%)$ | -19.21 | -39.43 | +4.68 | -29.03 | -15.44 | -8.53 |
| | $\text{Split-LoRA}_{(4\times8)}$ | Before | $50.68_{\pm1.06}$ | $93.08_{\pm0.41}$ | $77.12_{\pm0.76}$ | $48.65_{\pm1.18}$ | $79.30_{\pm0.91}$ | $86.05_{\pm0.44}$ |
| | | After | $41.83_{\pm1.92}$ | $59.37_{\pm0.59}$ | $81.70_{\pm0.52}$ | $22.98_{\pm1.35}$ | $67.02_{\pm0.59}$ | $81.01_{\pm1.34}$ |
| | | $\Delta\,(\%)$ | -8.85 | -33.71 | +4.58 | -25.67 | -12.28 | -5.04 |
| | $\text{FlyLoRA}_{(k=8)}$ | Before | $53.68_{\pm0.47}$ | $95.55_{\pm0.18}$ | $80.82_{\pm0.56}$ | $54.34_{\pm2.13}$ | $82.85_{\pm0.52}$ | $89.63_{\pm0.55}$ |
| | | After | $60.23_{\pm0.95}$ | $71.78_{\pm0.41}$ | $85.62_{\pm0.43}$ | $33.11_{\pm1.61}$ | $75.28_{\pm1.72}$ | $87.15_{\pm1.26}$ |
| | | $\Delta\,(\%)$ | **+6.55** | **-23.77** | **+4.80** | **-21.23** | **-7.57** | **-2.48** |

Table 3: Ablation study of FlyLoRA variants analyzing: **(a)** Load balancing in single-task (ST) setting, **(b)** Matrix $A$ freezing in both ST and multi-task merging (MT) settings, where LB=Load Balancing, Frz=$A$ Frozen, Trn=$A$ Trainable.

| Load Balancing | | Frozen $A$ | |
|---|---|---|---|
| Variant | Acc (%) | Variant | Acc (%) |
| w/ LB | **$40.88_{\pm1.61}$** | ST+Frz | **$40.88_{\pm1.61}$** |
| w/o LB | $37.56_{\pm2.87}$ | ST+Trn | $40.64_{\pm1.35}$ |
| | | MT+Frz | **$38.86_{\pm1.46}$** |
| | | MT+Trn | $34.43_{\pm2.24}$ |

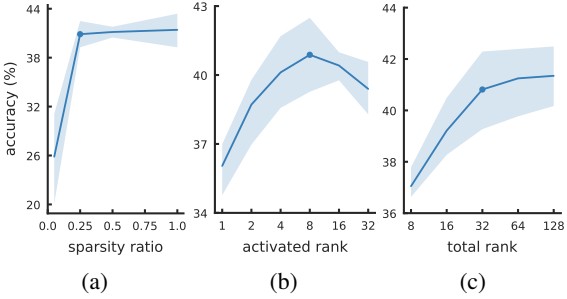

Figure 4: Accuracy comparison for: **(a)** Sparsity ratio in $A$, **(b)** Activated rank (with fixed total rank $r = 32$), **(c)** Total rank (with fixed activated rank $k = 8$).

updatable version, as shown in Table 3 (MMLU and Llama-3.1-8B). Our results demonstrate that load-balancing significantly improves MoE's training stability and boosts accuracy by 3.32%. For matrix $A$, we observe minimal performance differences between frozen and updatable versions in single-task settings. However, in multi-task merging scenarios, using an updatable matrix leads to a 4.43% performance degradation, as the updatable $A$ does not satisfy the approximately orthogonal property. More ablation studies are provided in Appendix B.

Further sensitivity analysis on single-task performance (Figure 4) reveals three key insights. First, model accuracy increases monotonically with the sparsity ratio of $A$ before saturating, exhibiting only marginal degradation unless the sparsity ratio becomes extremely small. Second, under a fixed total rank budget, performance peaks at intermediate activation ranks: insufficient rank fails to capture task-specific features, while excessive rank induces parameter interference. Third, increasing the total rank while holding the activation rank constant consistently yields performance gains.

# 5 Discussion

## 5.1 Interference in Model Merging

In scenarios requiring training, gradient orthogonalization techniques are commonly employed to reduce task interference, as seen in multi-task learning [88] and continual learning [8, 91]. In our training-free model merging setting, all components are derived from the same base model through domain-specific SFT. Task interference can be quantified by measuring the orthogonality of parameter updates (relative to the base model) across different tasks [29]. For our LoRA components merging, these parameter updates correspond to $B_i A_i$. We formally prove the near-orthogonality of FlyLoRA in Appendix A.3, which inherently reduces inter-task correlations.

## 5.2 FlyLoRA's Connection to Other Orthogonality-Based Designs in PEFT

Representative orthogonality-based PEFT methods like OFT [46, 59] and LoReFT [83] both operate on single tasks, and their orthogonal matrix $R$ multiplies the pre-trained weight matrix $W_0$, differing from LoRA variants (including FlyLoRA) that add $\Delta W$ to $W_0$. The multiplication scheme rotates the entire weight parameter space, and [46, 59] demonstrate this better adjusts semantic information compared to changing magnitude, explaining its success. In contrast, the additive scheme lacks this property since $W_0$ cannot be rotated. In single-task settings, removing the MoE part with only random $A$ reduces FlyLoRA to LoRA-FA [94] or Asymmetry LoRA [97]. These variants can save resources but cannot improve performance. Thus, although all methods use orthogonality, FlyLoRA succeeds differently. We think the orthogonality design in LoRA excels in *multi-task* scenarios, such as model merging (this work and LoRI [93]) and continual learning (O-LoRA [80]), because it decouples parameter interference across multiple downstream tasks when fine-tuning from the base model.

# 6 Related Work

**Low-Rank Adaptation** LoRA [27] is a widely used PEFT strategy for fine-tuning LLMs. To enhance its expressive power, several improvements [45, 95] have been proposed. Recently, to address parameter interference in settings like multi-task and continual learning, several MoE-based LoRA variants [17, 20, 21, 37, 55, 76, 82, 89] have proven effective by forcing each expert to specialize in specific areas. Several works [73, 75, 93] also aim to reduce interference during LoRA merging. In this article, we further develop the MoE-based LoRA structure. To improve LoRA's efficiency, LoRA-FA [94] and AsymmetryLoRA [97] show that fixing the down-projection matrix $A$ saves memory for input activations without performance degradation. In our work, we reconsider freezing $A$ from a new perspective by showing that its orthogonality and distance-preserving properties can be utilized to design an improved intra-/inter-task decoupling mechanism.

**Fly Olfactory Circuit** The fly olfactory circuit [6, 13, 38, 42, 72] serves as an exemplary model in bio-inspired AI due to its structural simplicity and functional completeness. Its core mechanism—random projection followed by sparse selection—effectively transforms high-dimensional inputs into separable representations. This biological principle has inspired algorithmic innovations across multiple AI domains, including locality-sensitive hashing [14, 67, 69], word embedding [41], federated learning [65], and continual learning [100, 101]. The circuit's enduring relevance highlights the value of cross-disciplinary inspiration in advancing computational methods.

# 7 Conclusion

In summary, this work provides a comprehensive revisit of the MoE-based structure for LoRA and analyzes its drawbacks regarding parameter interference and efficiency. Inspired by the fly olfactory circuit, we introduce FlyLoRA, a novel MoE-based LoRA variant that employs rank-wise expert activation in matrix $B$ and a fixed sparse random projection for matrix $A$ as an implicit router. Through the theoretical properties of these components, FlyLoRA achieves both intra-task and inter-task decoupling, significantly improving decorrelation in single-domain instruction tuning and LoRA component fusion in multi-task settings. Additionally, the implicit routing strategy and inherent sparsity ensure computational efficiency.

# 8 Acknowledgments

We thank Cheems Wang for his valuable suggestions on the manuscript. We also thank the anonymous reviewers for their positive feedback and constructive comments. This work was supported by the National Key R&D Program of China under Grant 2018AAA0102801.

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

# A Theoretical Analysis

## A.1 Distance Preserving Property for Sparse Random Projection

In this section, we prove that the fixed sparse random projection matrix $\boldsymbol{A}$ satisfies the distance-preserving property. This demonstrates that a fixed projection $\boldsymbol{A}$ can function like a hash router without an explicit router. Our result extends the well-known Johnson-Lindenstrauss Lemma [32]. First, [5, Corollary 3.5] provides the following bound:

**Theorem A.1.** *Let $\boldsymbol{A} \in \mathbb{R}^{r \times n}$ be a random matrix whose entries $\boldsymbol{A}_{ij}$ are sampled independently and randomly from a distribution that is symmetric around the origin with $\mathbb{E}(\boldsymbol{A}_{ij}^2) = \sigma^2 > 0$.*

1. *Suppose $C = \mathbb{E}(\boldsymbol{A}_{ij}^4) < \infty$. Then, for any $\epsilon > 0$,*

$$\mathbb{P}\left(\|\frac{1}{\sqrt{r}}\boldsymbol{A}\boldsymbol{x}\|^2 \leq \sigma^2(1-\epsilon)\|\boldsymbol{x}\|^2\right) \leq \exp\left(-\frac{(\epsilon^2-\epsilon^3)r}{2(\frac{1}{\sigma^4}C+1)}\right), \quad \textit{for all } \boldsymbol{x} \in \mathbb{R}^n.$$

2. *Suppose $\exists L > 0$ such that for any integer $k > 0$, $\mathbb{E}(\boldsymbol{A}_{ij}^{2k}) \leq \sigma^{2k}\frac{(2k)!}{2^k k!}L^{2k}$. Then, for any $\epsilon > 0$,*

$$\mathbb{P}\left(\|\frac{1}{\sqrt{r}}\boldsymbol{A}\boldsymbol{x}\|^2 \geq \sigma^2(1+\epsilon)L^2\|\boldsymbol{x}\|^2\right) \leq \exp\left(-(\epsilon^2-\epsilon^3)\frac{r}{4}\right), \quad \textit{for all } \boldsymbol{x} \in \mathbb{R}^n.$$

According to the definition of $\boldsymbol{A}$ mentioned in Section 3.1, the second moment of $\boldsymbol{A}_{ij}$ satisfies $\mathbb{E}(\boldsymbol{A}_{ij}^2) = \frac{p}{nr^2} > 0$. So, the $2k$-th moment of $\boldsymbol{A}_{ij}$ is given by $\mathbb{E}[\boldsymbol{A}_{ij}^{2k}] = \frac{p}{n} \cdot \mathbb{E}[\boldsymbol{X}_{ij}^{2k}] + \left(1 - \frac{p}{n}\right) \cdot 0 = \frac{p}{n} \cdot \mathbb{E}[\boldsymbol{X}_{ij}^{2k}]$, where $\boldsymbol{X}_{ij} \sim \mathcal{N}(0, \frac{1}{r^2})$. From the property of Gaussian distribution, we derive $\mathbb{E}[\boldsymbol{X}_{ij}^{2k}] = (2k-1)!! \cdot \left(\frac{1}{r^2}\right)^{2k} = \frac{(2k)!}{2^k k!}\left(\frac{1}{r^2}\right)^{2k}$, where !! denotes the double factorial. This leads to $\mathbb{E}[\boldsymbol{A}_{ij}^4] = \frac{3p}{nr^4}$, which is clearly finite. To satisfy the inequality $\mathbb{E}[\boldsymbol{A}_{ij}^{2k}] = \frac{(2k)!}{2^k k!}\sigma^{2k} \leq \sigma^{2k}\frac{(2k)!}{2^k k!}L^{2k}$, we require $L \geq \log_{2k}\frac{p}{r}$. Due to monotonicity, simply choosing $L = \log_2 \frac{p}{r}$ always satisfies the condition specified in Theorem A.1. Combining both bounds from Theorem A.1, we summarize the final result in Theorem A.2.

$$\mathbb{P}\left((1-\epsilon)\|\boldsymbol{x}-\boldsymbol{y}\|^2 \leq \frac{1}{r\sigma^2}\|\boldsymbol{A}\boldsymbol{x}-\boldsymbol{A}\boldsymbol{y}\|^2 \leq (1+\epsilon)\|\boldsymbol{x}-\boldsymbol{y}\|^2\right) =$$
$$1 - \mathbb{P}\left(\frac{1}{r}\|\boldsymbol{A}\boldsymbol{x}-\boldsymbol{A}\boldsymbol{y}\|^2 < (1-\epsilon)\sigma^2\|\boldsymbol{x}-\boldsymbol{y}\|^2\right) - \tag{13}$$
$$\mathbb{P}\left(\frac{1}{r}\|\boldsymbol{A}\boldsymbol{x}-\boldsymbol{A}\boldsymbol{y}\|^2 > (1+\epsilon)\sigma^2\|\boldsymbol{x}-\boldsymbol{y}\|^2\right),$$

**Theorem A.2.** *Given the matrix $\boldsymbol{A} \in \mathbb{R}^{r \times n}$ with each entry i.i.d. from $\mathcal{N}(0, \frac{1}{r^2})$, and set to $0$ otherwise, for any $\epsilon > 0$,*

$$\mathbb{P}\left((1-\epsilon)\|\boldsymbol{x}-\boldsymbol{y}\|^2 \leq \frac{1}{r\sigma^2}\|\boldsymbol{A}\boldsymbol{x}-\boldsymbol{A}\boldsymbol{y}\|^2 \leq (1+\epsilon)\|\boldsymbol{x}-\boldsymbol{y}\|^2\right) \geq 1 - e^{-(\epsilon^2-\epsilon^3)\frac{r}{4}} - e^{-\frac{(\epsilon^2-\epsilon^3)r}{2(\frac{3p}{n}+1)}},$$

*for any input embeddings $\boldsymbol{x}, \boldsymbol{y} \in \mathbb{R}^n$, where $\sigma^2 = \frac{p}{nr^2}$.*

Since this construction does not differ significantly from our desired construction for $\boldsymbol{A}$, which includes an additional constraint on non-zero entries per row and is more consistent with biological observations in the fly olfactory circuit, we use this more analytically tractable form as a surrogate to study the distance-preserving property. In practice, these two construction methods show negligible performance differences.

## A.2 Top-k Activation Promotes Rank-Wise Decoupling

Let $\boldsymbol{\Lambda} \in \text{diag}(\lambda_1, \ldots, \lambda_r) \in \mathbb{R}^{r \times r}$ denote a binary mask where $\lambda_i = 1$ if column $i$ is activated by top-$k$. The relation of the masked gradient between top-$k$ version ($\frac{\partial\mathcal{L}}{\partial\tilde{\boldsymbol{B}}}$) and the dense version ($\frac{\partial\mathcal{L}}{\partial\boldsymbol{B}}$) is:

$$\frac{\partial\mathcal{L}}{\partial\tilde{\boldsymbol{B}}} = \frac{\partial\mathcal{L}}{\partial\boldsymbol{B}}\boldsymbol{\Lambda} \in \mathbb{R}^{m \times r}. \tag{14}$$

Let $\tilde{g}_i$ and $g_i \in \mathbb{R}^m$ denote the gradient vectors for the $i$-th column of $\frac{\partial \mathcal{L}}{\partial \tilde{B}}$ and $\frac{\partial \mathcal{L}}{\partial B}$, respectively. Their corresponding cross-column covariance are:

$$\Sigma_{(i,j)} = \mathbb{E}\left[g_i^\top g_j\right] - \mathbb{E}\left[g_i\right]^\top \mathbb{E}\left[g_j\right], \tag{15}$$

$$\tilde{\Sigma}_{(i,j)} = \mathbb{E}\left[\tilde{g}_i^\top \tilde{g}_j\right] - \mathbb{E}\left[\tilde{g}_i\right]^\top \mathbb{E}\left[\tilde{g}_j\right], \tag{16}$$

We can assume $\mathbb{E}[\tilde{g}_i] = \mathbb{E}[\tilde{g}_j] = \mathbb{E}[g_i] = \mathbb{E}[g_j] = \mathbf{0}_m$, these simplify to:

$$\Sigma_{(i,j)} = \mathbb{E}\left[g_i^\top g_j\right], \tag{17}$$

$$\tilde{\Sigma}_{(i,j)} = \mathbb{E}\left[\tilde{g}_i^\top \tilde{g}_j\right]. \tag{18}$$

Thus, the expected covariance depends only on the expected inner product of gradient vectors. From Assumption 3.2, the difference between $\tilde{\Sigma}_{(i,j)}, \Sigma_{(i,j)}$ is factored by the co-activation probability of columns $i$ and $j$:

$$\mathbb{P}(\lambda_i = 1 \cap \lambda_j = 1) = \frac{\binom{r-2}{k-2}}{\binom{r}{k}} = \frac{k(k-1)}{r(r-1)} \approx \frac{k^2}{r^2}. \tag{19}$$

This leads to the following theorem:

**Theorem A.3** (Covariance Reduction Under top-$k$). *Let $\tilde{\Sigma}$ and $\Sigma$ denote the gradient covariance matrices with and without top-$k$ activation. When $r > k$, the off-diagonal entries scale as:*

$$\mathbb{E}[\tilde{\Sigma}_{(i,j)}] \approx \mathbb{E}[\Sigma_{(i,j)}] \cdot \frac{k^2}{r^2}, \quad \forall i \neq j. \tag{20}$$

### A.3 Random Projection Induces Approximate Subspace Orthogonality

The orthogonality properties of random projections form the theoretical foundation for FlyLoRA's effectiveness in model merging. Consider two independent random projection matrices $A_i, A_j \in \mathbb{R}^{r \times n}$ with entries distributed as specified in Section 3.1. The expectation of their product reveals:

$$\mathbb{E}[A_i A_j^\top] = \mathbb{E}\left[\sum_{k=1}^n (A_i)_{mk}(A_j)_{lk}\right] = \sum_{k=1}^n \mathbb{E}[(A_i)_{mk}]\mathbb{E}[(A_j)_{lk}] = \mathbf{0}_{r \times r} \tag{21}$$

This zero-expectation result follows from the independence and zero-mean property of the random matrices. To quantify how tightly the product concentrates around zero, we analyze its variance:

$$\begin{aligned}
\mathrm{Var}\left((A_i A_j^T)_{ml}\right) &= \sum_{k=1}^n \Big[\mathrm{Var}\big((A_i)_{mk}\big)\mathrm{Var}\big((A_j)_{lk}\big) + \mathrm{Var}\big((A_i)_{mk}\big)\big(\mathbb{E}(A_j)_{lk}\big)^2 \\
&\quad + \mathrm{Var}\big((A_j)_{lk}\big)\big(\mathbb{E}(A_i)_{mk}\big)^2\Big] \\
&= \sum_{k=1}^n \mathrm{Var}\big((A_i)_{mk}\big)\mathrm{Var}\big((A_j)_{lk}\big) \\
&= n\sigma^4 = \frac{p^2}{nr^4}
\end{aligned} \tag{22}$$

Applying Chebyshev's inequality, we bound the probability of large deviations for each entry:

$$\mathbb{P}\left(|(A_i A_j^\top)_{ml}| \geq \epsilon\right) \leq \frac{\mathrm{Var}\left((A_i A_j^\top)_{ml}\right)}{\epsilon^2} = \frac{p^2}{nr^4\epsilon^2} \tag{23}$$

The Frobenius norm characterization gives us: $\|A_i A_j^\top\|_F^2 = \sum_{m=1}^r \sum_{l=1}^r (A_i A_j^\top)_{ml}^2$. To analyze its probabilistic behavior, we employ the union bound principle: for any events $E_{ij}$ defined as $|(A_i A_j^\top)_{ml}| \geq \epsilon$, we have $\mathbb{P}\left(\bigcup_{m,l} E_{ml}\right) \leq \sum_{m,l} \mathbb{P}(E_{ml})$. Using this union bound over all $r^2$ entries yields:

$$\mathbb{P}\left(\|A_i A_j^\top\|_F \geq \epsilon r\right) \leq \sum_{m=1}^r \sum_{l=1}^r \mathbb{P}\left(|(A_i A_j^\top)_{ml}| \geq \epsilon\right) \leq r^2 \cdot \frac{p^2}{nr^4\epsilon^2} = \frac{p^2}{nr^2\epsilon^2}. \tag{24}$$

Since the spectral norm is bounded by the Frobenius norm ($\|A_i^\top A_j\|_2 \leq \|A_i^\top A_j\|_F$), we conclude that when $n \gg \frac{p^2}{r^2\epsilon^2}$, the subspaces spanned by $A_i$ and $A_j$ are approximately orthogonal with high probability.

**Theorem A.4** (Approximate Subspace Orthogonality). *For independent random matrices $\boldsymbol{A}_i, \boldsymbol{A}_j \in \mathbb{R}^{r \times n}$ with sparse Gaussian entries ($\mathcal{N}(0, \frac{1}{r^2})$ for $p < n$ randomly selected entries per row), the following holds:*

1. **Exact mean orthogonality**: $\mathbb{E}[\boldsymbol{A}_i \boldsymbol{A}_j^\top] = \mathbf{0}_{r \times r}$

2. **Polynomially decaying correlations**: $\mathbb{P}(\|\boldsymbol{A}_i \boldsymbol{A}_j^\top\|_2 \geq \epsilon r) \leq \frac{p^2}{nr^2\epsilon^2}$

We demonstrate the approximate orthogonality between distinct LoRA components $\boldsymbol{B}_i \boldsymbol{A}_i$ and $\boldsymbol{B}_j \boldsymbol{A}_j$ through Frobenius inner product analysis following [75]:

$$
\begin{aligned}
\langle \boldsymbol{B}_j \boldsymbol{A}_j, \boldsymbol{B}_i \boldsymbol{A}_i \rangle_F &= \mathrm{tr}\left( (\boldsymbol{B}_j \boldsymbol{A}_j)^\top (\boldsymbol{B}_i \boldsymbol{A}_i) \right) \\
&= \mathrm{tr}\left( \boldsymbol{A}_j^\top \boldsymbol{B}_j^\top \boldsymbol{B}_i \boldsymbol{A}_i \right) \\
&= \mathrm{tr}\left( \boldsymbol{B}_j^\top \boldsymbol{B}_i \boldsymbol{A}_i \boldsymbol{A}_j^\top \right) \\
&\approx \mathrm{tr}\left( \boldsymbol{B}_j^\top \boldsymbol{B}_i \cdot \mathbf{0}_{r \times r} \right) \\
&\approx 0
\end{aligned}
\tag{25}
$$

These analysis demonstrates that random projections naturally create nearly orthogonal subspaces, which helps prevent interference between different experts in FlyLoRA. The small residual correlation becomes negligible when input dimension $n \to \infty$, leading to the effectiveness of our sparse random projection approach for model merging.

Formally, considering the conventional LoRA merging scheme:

$$
\boldsymbol{W}' = \boldsymbol{W}_0 + \sum_{i=1}^{t} w_i \boldsymbol{B}_i \boldsymbol{A}_i
\tag{26}
$$

When multiple FlyLoRA modules ($\boldsymbol{B}_i \boldsymbol{A}_i$) are approximately orthogonal, the squared Frobenius norm of the merged weight matrix can be decomposed into the weighted sum of individual module norms.

$$
\left\| \sum_{i=1}^{t} w_i \boldsymbol{B}_i \boldsymbol{A}_i \right\|_F^2 = \sum_{i=1}^{t} w_i^2 \|\boldsymbol{B}_i \boldsymbol{A}_i\|_F^2 + \sum_{i \neq j} w_i w_j \langle \boldsymbol{B}_i \boldsymbol{A}_i, \boldsymbol{B}_j \boldsymbol{A}_j \rangle_F
\tag{27}
$$

$$
\approx \sum_{i=1}^{t} w_i^2 \|\boldsymbol{B}_i \boldsymbol{A}_i\|_F^2,
\tag{28}
$$

which aligns with the "Weight disentanglement" property for task arithmetic [56]. We summarize these findings in the following corollary:

**Corollary A.5.** *Let $\boldsymbol{A}_i, \boldsymbol{A}_j \in \mathbb{R}^{r \times n}$ be fixed sparse random projections after initialization. Then for any learned matrices $\boldsymbol{B}_i \boldsymbol{A}_i$ and $\boldsymbol{B}_j \boldsymbol{A}_j$ the following properties hold:*

1. **Pairwise Orthogonality**:

$$
\langle \boldsymbol{B}_i \boldsymbol{A}_i, \boldsymbol{B}_j \boldsymbol{A}_j \rangle_F \approx 0 \quad for \quad i \neq j
$$

2. **Orthogonality's Outcome in Merging**:

$$
\left\| \sum_{i=1}^{t} w_i \boldsymbol{B}_i \boldsymbol{A}_i \right\|_F^2 \approx \sum_{i=1}^{t} w_i^2 \|\boldsymbol{B}_i \boldsymbol{A}_i\|_F^2
$$

Through sparse random projections, FlyLoRA inherently constructs nearly orthogonal subspaces for parameter merging from different tasks.

# B  Additional Results

## B.1  Evaluation on Larger Models

We further conducted experiments using the Qwen-2.5-14B model, as shown in Tables 4 and 5, following the settings of Tables 1 and 2. The results show that FlyLoRA remains superior in accuracy

(for both single-task and multi-task settings) and is more parameter-efficient. We encountered no memory or convergence bottlenecks when training FlyLoRA on the 14B model, which consistently outperforms LoRA and Split-LoRA. This scalability confirms that our method's benefits extend effectively to larger model architectures without compromising training stability.

Table 4: **Performance Comparison of LoRA Variants in Single-task Evaluation using Qwen-2.5-14B.** We evaluate various methods across four benchmarks: MMLU, ScienceQA, GSM8K (accuracy), and HumanEval (Pass@k), with all metrics reported in percentage (%). Param(%) indicates the percentage of activated trainable parameters relative to Full FT. The best results are highlighted in **bold**.

| Method | Param(%) | MMLU | ScienceQA | GSM8K | HumanEval |
|---|---|---|---|---|---|
| LoRA$_{(r=8)}$ | 0.23 | $56.74_{\pm0.56}$ | $95.62_{\pm0.18}$ | $83.08_{\pm0.74}$ | $51.69_{\pm1.48}$ |
| LoRA$_{(r=32)}$ | 0.93 | $59.35_{\pm0.79}$ | $97.05_{\pm0.22}$ | $85.31_{\pm0.25}$ | $54.80_{\pm0.76}$ |
| Split-LoRA$_{(4\times8)}$ | 0.29 | $58.26_{\pm1.13}$ | $96.85_{\pm0.35}$ | $84.88_{\pm0.51}$ | $54.65_{\pm0.59}$ |
| FlyLoRA | **0.12** | $\mathbf{60.17_{\pm1.08}}$ | $\mathbf{97.37_{\pm0.32}}$ | $\mathbf{85.96_{\pm0.89}}$ | $\mathbf{56.42_{\pm1.16}}$ |

Table 5: **Multi-task Performance Comparison using Qwen-2.5-14B.** We evaluate LoRA variants across MMLU, ScienceQA, GSM8K (accuracy), and HumanEval (Pass@k) benchmarks. The table shows the relative performance drop ($\Delta\%$) before and after merging. The best results are highlighted in **bold**.

| Method | MMLU | ScienceQA | GSM8K | HumanEval |
|---|---|---|---|---|
| LoRA$_{(r=8)}$ | -13.75 | -25.20 | -11.43 | -18.60 |
| LoRA$_{(r=32)}$ | -8.91 | -20.45 | -7.62 | -16.34 |
| Split-LoRA$_{(4\times8)}$ | -7.48 | -21.97 | -6.05 | -14.87 |
| FlyLoRA | **-4.35** | **-17.89** | **-2.18** | **-11.72** |

## B.2 More Baseline into Comparison

We further compare several strong and widely used baselines—AdaLoRA [95] (adaptive rank allocation), SoRA [16] (sparse adaptation), and HydraLoRA [76] (MoE-based)—using Qwen-2.5-7B, as shown in Tables 6 and 7. The results demonstrate that FlyLoRA remains superior in terms of accuracy and efficiency in both single-task learning and multi-task merging scenarios. Notably, FlyLoRA achieves this performance while maintaining a simpler training pipeline, as it avoids the additional hyperparameter tuning required by adaptive and sparse methods.

Table 6: **Performance Comparison with More LoRA Variants in Single-task Evaluation using Qwen-2.5-7B.** We evaluate various methods across four benchmarks: MMLU, ScienceQA, GSM8K (accuracy), and HumanEval (Pass@k), with all metrics reported in percentage (%). Param(%) indicates the percentage of activated trainable parameters relative to Full FT. The best results are highlighted in **bold**.

| Method | Param(%) | MMLU | ScienceQA | GSM8K | HumanEval |
|---|---|---|---|---|---|
| AdaLoRA$_{(r=8)}$ | 0.26 | $51.22_{\pm0.21}$ | $93.48_{\pm0.28}$ | $77.65_{\pm0.14}$ | $47.96_{\pm1.34}$ |
| SoRA$_{(r=8)}$ | 0.19 | $50.89_{\pm0.42}$ | $93.25_{\pm0.20}$ | $78.46_{\pm0.82}$ | $47.83_{\pm0.94}$ |
| HydraLoRA$_{(r=8,A=1,B=3)}$ | 0.52 | $53.05_{\pm0.16}$ | $94.69_{\pm0.34}$ | $79.31_{\pm0.49}$ | $52.98_{\pm1.57}$ |
| FlyLoRA$_{(k=8)}$ | **0.13** | $\mathbf{53.68_{\pm0.47}}$ | $\mathbf{95.55_{\pm0.18}}$ | $\mathbf{80.82_{\pm0.56}}$ | $\mathbf{54.34_{\pm2.13}}$ |

## B.3 Training Time and Memory Consumption

We further conduct experiments comparing the training time and memory consumption of the LoRA variants discussed in Section 4. The results, presented in Table 8, demonstrate that LoRA$_{(r=32)}$ requires more training time and memory than LoRA$_{(r=8)}$ due to its higher rank. Split-LoRA$_{(4\times8)}$ shows intermediate values between these two approaches. Notably, FlyLoRA achieves both the fastest training times (resulting from having the fewest activated parameters) and the lowest memory consumption (mainly attributed to its frozen matrix $A$ that significantly reduces memory for activation

Table 7: **Multi-task Performance Comparison with More Baselines using Qwen-2.5-7B.** We evaluate LoRA variants across MMLU, ScienceQA, GSM8K (accuracy), and HumanEval (Pass@k) benchmarks. The table shows the relative performance drop ($\Delta\%$) before and after merging. The best results are highlighted in **bold**.

| Method | MMLU | ScienceQA | GSM8K | HumanEval |
|---|---|---|---|---|
| AdaLoRA$_{(r=8)}$ | -10.90 | -33.15 | +4.52 | -25.46 |
| SoRA$_{(r=8)}$ | -9.45 | -34.81 | +4.05 | -26.50 |
| HydraLoRA$_{(r=8,A=1,B=3)}$ | -6.22 | -34.67 | +4.21 | -24.94 |
| FlyLoRA$_{(k=8)}$ | **+6.55** | **-23.77** | **+4.80** | **-21.23** |

values [94]). We also illustrate the theoretical memory consumption for these LoRA variants in Table 9, which tightly aligns with the experimental results. It's clear to see that under a fixed total rank $r$, a larger number of experts $N$ causes Split-LoRA to require significantly more activated trainable parameters and memory consumption, degrading the efficiency of MoE-based LoRA methods. These results and analysis demonstrate the robustness of FlyLoRA's parameter efficiency compared to MoE-based LoRA methods.

Table 8: **Training Time (Hours) and Memory Usage (GB) of LoRA Variants on Different Datasets and Architectures.** Comparison of LoRA$_{(r=8)}$, LoRA$_{(r=32)}$, Split-LoRA$_{(4\times8)}$, and FlyLoRA$_{(k=8)}$ fine-tuning Llama-3.1-8B and Qwen-2.5-7B on MMLU, ScienceQA, GSM8K, and CodeAlpaca-20k. The best results are highlighted in **bold**.

| Metric | Llama-3.1-8B | | | | Qwen-2.5-7B | | | |
|---|---|---|---|---|---|---|---|---|
| | MMLU | ScienceQA | GSM8K | CodeAlpaca | MMLU | ScienceQA | GSM8K | CodeAlpaca |
| LoRA$_{(r=8)}$ | | | | | | | | |
| Training Time | 4.79h | 5.30h | 0.52h | 1.85h | 4.45h | 5.07h | 0.54h | 1.75h |
| Memory Usage | 12.5GB | 14.8GB | 20.1GB | 20.2GB | 14.7GB | 16.8GB | 22.9GB | 22.9GB |
| LoRA$_{(r=32)}$ | | | | | | | | |
| Training Time | 5.09h | 5.61h | 0.58h | 1.95h | 4.76h | 5.39h | 0.60h | 1.87h |
| Memory Usage | 13.2GB | 15.3GB | 20.7GB | 20.7GB | 15.4GB | 17.3GB | 23.4GB | 23.4GB |
| Split-LoRA$_{(4\times8)}$ | | | | | | | | |
| Training Time | 4.94h | 5.46h | 0.56h | 1.92h | 4.60h | 5.23h | 0.57h | 1.79h |
| Memory Usage | 12.8GB | 15.0GB | 20.4GB | 20.4GB | 15.0GB | 17.1GB | 23.2GB | 23.2GB |
| FlyLoRA$_{(k=8)}$ | | | | | | | | |
| Training Time | **4.73h** | **5.23h** | **0.51h** | **1.82h** | **4.39h** | **4.99h** | **0.52h** | **1.70h** |
| Memory Usage | **10.6GB** | **10.7GB** | **10.9GB** | **10.9GB** | **12.1GB** | **12.2GB** | **12.4GB** | **12.4GB** |

Table 9: **Theoretical Memory Consumption Comparison of Different LoRA Variants for a Single Linear Layer.** Param indicates the number of activated trainable parameters. Variables $d$, $r$, $k$, $b$, $s$, and $N$ represent hidden dimension, total rank, activation rank, batch size, sequence length, and number of experts, respectively. We record memory usage for weights, gradients, optimizer states, and activations in bytes. These results are calculated under 16-bit mixed-precision training settings.

| Method | Param | Weight | Gradient | Optimizer | Activation |
|---|---|---|---|---|---|
| LoRA | $2dr$ | $2(d^2 + 2dr)$ | $4dr$ | $24dr$ | $2bsd + 2bsr$ |
| Split-LoRA | $2dk + dN$ | $2(d^2 + 2dr + dN)$ | $4dk + 2dN$ | $24dk + 12dN$ | $2bsd + 2bsk + 2bsN$ |
| FlyLoRA | $dk$ | $2(d^2 + 2dr)$ | $2dk$ | $12dk$ | $2bsk$ |

## B.4 Multi-task Performance for Advanced Model Merging Techniques

Extending the results in Section 4.3, we also evaluate two advanced model merging techniques, TIES-MERGING [85] and DARE [87], for merging LoRA components from different domains. The results are listed in Tables 10 and 11, respectively. Overall, both TIES-MERGING and DARE outperform naive weight averaging by resolving parameter conflicts through intelligent selection (TIES' sign consensus and trimming) and selective rescaling (DARE's dropout-based redundancy elimination). Similar to weight averaging, these results demonstrate that FlyLoRA consistently surpasses LoRA$_{(r=8)}$, LoRA$_{(r=32)}$, and Split-LoRA$_{(4\times8)}$ across all comparisons, highlighting the

Table 10: **Multi-task Performance Comparison Before and After Parameter Merging Using TIES-MERGING.** We evaluate LoRA variants across MMLU, ScienceQA, GSM8K (accuracy), and HumanEval (Pass@k) benchmarks. The table shows performance before merging, after merging, and the relative performance drop ($\Delta\%$). The best results are highlighted in **bold**.

| Model | Method | Merge Status | MMLU | ScienceQA | GSM8K | HumanEval Pass@1 | Pass@5 | Pass@10 |
|---|---|---|---|---|---|---|---|---|
| Llama-3.1-8B | $\text{LoRA}_{(r=8)}$ | Before | $36.53_{\pm0.40}$ | $91.39_{\pm0.55}$ | $55.34_{\pm0.24}$ | $29.13_{\pm0.56}$ | $52.28_{\pm1.24}$ | $61.67_{\pm0.61}$ |
| | | After | $31.83_{\pm0.34}$ | $35.96_{\pm1.46}$ | $27.53_{\pm1.08}$ | $18.45_{\pm1.47}$ | $45.52_{\pm0.84}$ | $56.63_{\pm1.24}$ |
| | | $\Delta$ (%) | -4.70 | -55.43 | -27.81 | -10.68 | -6.76 | -5.04 |
| | $\text{LoRA}_{(r=32)}$ | Before | $38.93_{\pm1.04}$ | $94.01_{\pm0.17}$ | $56.25_{\pm0.29}$ | $30.37_{\pm1.06}$ | $54.37_{\pm0.39}$ | $64.02_{\pm0.94}$ |
| | | After | $34.45_{\pm1.73}$ | $36.63_{\pm0.79}$ | $26.59_{\pm0.47}$ | $20.98_{\pm1.13}$ | $47.15_{\pm0.82}$ | $59.97_{\pm1.04}$ |
| | | $\Delta$ (%) | -4.48 | -57.38 | -29.66 | -9.39 | -7.22 | -4.05 |
| | $\text{Split-LoRA}_{(4\times8)}$ | Before | $38.44_{\pm0.69}$ | $92.41_{\pm0.54}$ | $55.65_{\pm0.47}$ | $31.28_{\pm1.52}$ | $54.16_{\pm1.12}$ | $63.94_{\pm0.89}$ |
| | | After | $33.92_{\pm0.57}$ | $40.81_{\pm1.34}$ | $29.02_{\pm0.82}$ | $22.02_{\pm0.24}$ | $46.32_{\pm0.72}$ | $59.72_{\pm0.25}$ |
| | | $\Delta$ (%) | -4.52 | -51.60 | -26.63 | -9.26 | -7.84 | -4.22 |
| | $\text{FlyLoRA}_{(k=8)}$ | Before | $40.88_{\pm1.61}$ | $94.15_{\pm0.36}$ | $58.76_{\pm0.74}$ | $36.88_{\pm1.91}$ | $62.40_{\pm1.82}$ | $73.34_{\pm1.24}$ |
| | | After | $39.03_{\pm1.31}$ | $54.82_{\pm0.46}$ | $39.12_{\pm1.02}$ | $33.37_{\pm0.62}$ | $57.36_{\pm1.41}$ | $70.35_{\pm0.39}$ |
| | | $\Delta$ (%) | **-1.85** | **-39.33** | **-19.64** | **-3.51** | **-5.04** | **-2.99** |
| Qwen-2.5-7B | $\text{LoRA}_{(r=8)}$ | Before | $49.84_{\pm0.56}$ | $92.84_{\pm0.13}$ | $77.01_{\pm0.32}$ | $47.20_{\pm1.54}$ | $78.89_{\pm0.36}$ | $85.94_{\pm0.64}$ |
| | | After | $44.98_{\pm0.72}$ | $61.69_{\pm1.34}$ | $81.89_{\pm1.04}$ | $23.37_{\pm1.24}$ | $68.96_{\pm1.22}$ | $81.25_{\pm0.35}$ |
| | | $\Delta$ (%) | -4.86 | -31.15 | +4.88 | -23.83 | -9.93 | -4.69 |
| | $\text{LoRA}_{(r=32)}$ | Before | $52.07_{\pm0.31}$ | $95.01_{\pm0.21}$ | $79.23_{\pm0.22}$ | $52.87_{\pm1.79}$ | $81.67_{\pm1.14}$ | $87.80_{\pm0.72}$ |
| | | After | $35.92_{\pm0.18}$ | $58.02_{\pm0.27}$ | $83.98_{\pm0.62}$ | $24.79_{\pm1.08}$ | $67.50_{\pm1.13}$ | $80.35_{\pm1.46}$ |
| | | $\Delta$ (%) | -16.15 | -36.99 | +4.75 | -28.08 | -14.17 | -7.45 |
| | $\text{Split-LoRA}_{(4\times8)}$ | Before | $50.68_{\pm1.06}$ | $93.08_{\pm0.41}$ | $77.12_{\pm0.76}$ | $48.65_{\pm1.18}$ | $79.30_{\pm0.91}$ | $86.05_{\pm0.44}$ |
| | | After | $44.96_{\pm0.65}$ | $60.59_{\pm0.26}$ | $81.82_{\pm0.20}$ | $23.53_{\pm0.74}$ | $67.59_{\pm0.14}$ | $82.34_{\pm0.69}$ |
| | | $\Delta$ (%) | -5.72 | -32.49 | +4.70 | -25.12 | -11.71 | -3.71 |
| | $\text{FlyLoRA}_{(k=8)}$ | Before | $53.68_{\pm0.47}$ | $95.55_{\pm0.18}$ | $80.82_{\pm0.56}$ | $54.34_{\pm2.13}$ | $82.85_{\pm0.52}$ | $89.63_{\pm0.55}$ |
| | | After | $60.51_{\pm0.37}$ | $73.46_{\pm0.80}$ | $86.24_{\pm0.31}$ | $35.37_{\pm0.47}$ | $76.05_{\pm1.25}$ | $87.97_{\pm1.09}$ |
| | | $\Delta$ (%) | **+6.83** | **-22.09** | **+5.42** | **-18.97** | **-6.80** | **-1.66** |

robustness of FlyLoRA's near-orthogonality property in reducing inter-task decoupling and further enhancing model merging performance.

We also include comparison with more advanced model merging techniques in Table 12. KnOTS [73] and L-LoRA [75] are both built upon $\text{LoRA}_{(r=32)}$. The results suggest that they achieve comparable performance to FlyLoRA, with each method excelling on different datasets. It is noteworthy that FlyLoRA is not a competitor to these methods; rather, they can be used in a plug-and-play manner with FlyLoRA to further improve performance after merging.

## B.5 Additional Ablation Studies on Load-Balancing Strategies

We compare experimental results using different load-balancing strategies. In Section 3.1, we employ an easy-to-implement loss-free balancing strategy. This loss-agnostic approach effectively achieves load balancing with negligible computational overhead and memory footprint. Simultaneously, other loss-controlled load-balancing strategies like [19] are widely used in MoE-like structures. A comparison of them is shown in Table 13. Our results show that different routing strategies achieve similar effects in performance. While FlyLoRA requires load-balancing strategies, it is not sensitive to specific methods.

## B.6 Additional Ablation Studies on K-Selection Strategies

To evaluate the impact of activation selection, we analyze different K-selection approaches in our experiments. In neuroscience, the fly olfactory circuit implements a "winner-take-all" strategy, simulated through top-$k$ selection based on activation values across dimensions. To validate top-$k$'s effectiveness, we test random-$k$ selection and full activation (without selection) as baselines. Results in Table 14 show that both top-$k$ and random-$k$ outperform full activation, confirming sparse activation mitigates intra-task interference. Crucially, top-$k$ surpasses random-$k$ because random selection cannot prioritize the most informative dimensions, leading to suboptimal performance.

Table 11: **Multi-task Performance Comparison Before and After Parameter Merging Using DARE.** We evaluate LoRA variants across MMLU, ScienceQA, GSM8K (accuracy), and HumanEval (Pass@k) benchmarks. The table shows performance before merging, after merging, and the relative performance drop ($\Delta\%$). The best results are highlighted in **bold**.

| Model | Method | Merge Status | MMLU | ScienceQA | GSM8K | HumanEval Pass@1 | Pass@5 | Pass@10 |
|---|---|---|---|---|---|---|---|---|
| Llama-3.1-8B | $\text{LoRA}_{(r=8)}$ | Before | $36.53_{\pm0.40}$ | $91.39_{\pm0.55}$ | $55.34_{\pm0.24}$ | $29.13_{\pm0.56}$ | $52.28_{\pm1.24}$ | $61.67_{\pm0.61}$ |
| | | After | $31.24_{\pm0.40}$ | $34.37_{\pm1.25}$ | $26.76_{\pm1.10}$ | $17.35_{\pm1.32}$ | $45.49_{\pm0.96}$ | $57.24_{\pm1.36}$ |
| | | $\Delta$ (%) | -5.29 | -57.02 | -28.58 | -11.78 | -6.79 | -4.43 |
| | $\text{LoRA}_{(r=32)}$ | Before | $38.93_{\pm1.04}$ | $94.01_{\pm0.17}$ | $56.25_{\pm0.29}$ | $30.37_{\pm1.06}$ | $54.37_{\pm0.39}$ | $64.02_{\pm0.94}$ |
| | | After | $34.75_{\pm0.83}$ | $37.56_{\pm0.59}$ | $26.89_{\pm0.78}$ | $19.36_{\pm1.38}$ | $46.67_{\pm0.86}$ | $59.85_{\pm0.67}$ |
| | | $\Delta$ (%) | -4.18 | -56.45 | -29.36 | -11.01 | -7.70 | -4.17 |
| | $\text{Split-LoRA}_{(4\times8)}$ | Before | $38.44_{\pm0.69}$ | $92.41_{\pm0.54}$ | $55.65_{\pm0.47}$ | $31.28_{\pm1.52}$ | $54.16_{\pm1.12}$ | $63.94_{\pm0.89}$ |
| | | After | $34.02_{\pm0.24}$ | $38.58_{\pm0.48}$ | $28.63_{\pm0.72}$ | $23.52_{\pm1.43}$ | $46.84_{\pm0.30}$ | $60.30_{\pm0.41}$ |
| | | $\Delta$ (%) | -4.42 | -53.83 | -27.02 | -7.76 | -7.32 | -3.64 |
| | $\text{FlyLoRA}_{(k=8)}$ | Before | $40.88_{\pm1.61}$ | $94.15_{\pm0.36}$ | $58.76_{\pm0.74}$ | $36.88_{\pm1.91}$ | $62.40_{\pm1.82}$ | $73.34_{\pm1.24}$ |
| | | After | $39.37_{\pm1.02}$ | $52.34_{\pm0.35}$ | $38.20_{\pm1.52}$ | $33.34_{\pm0.83}$ | $57.14_{\pm1.37}$ | $70.24_{\pm0.42}$ |
| | | $\Delta$ (%) | **-1.51** | **-41.81** | **-20.56** | **-3.54** | **-5.26** | **-3.10** |
| Qwen-2.5-7B | $\text{LoRA}_{(r=8)}$ | Before | $49.84_{\pm0.56}$ | $92.84_{\pm0.13}$ | $77.01_{\pm0.32}$ | $47.20_{\pm1.54}$ | $78.89_{\pm0.36}$ | $85.94_{\pm0.64}$ |
| | | After | $45.20_{\pm0.40}$ | $61.39_{\pm1.32}$ | $81.98_{\pm0.86}$ | $23.49_{\pm1.02}$ | $69.04_{\pm0.17}$ | $81.22_{\pm0.27}$ |
| | | $\Delta$ (%) | -4.64 | -31.45 | +4.97 | -23.71 | -9.85 | -4.72 |
| | $\text{LoRA}_{(r=32)}$ | Before | $52.07_{\pm0.31}$ | $95.01_{\pm0.21}$ | $79.23_{\pm0.22}$ | $52.87_{\pm1.79}$ | $81.67_{\pm1.14}$ | $87.80_{\pm0.72}$ |
| | | After | $35.27_{\pm1.68}$ | $56.79_{\pm1.34}$ | $84.07_{\pm0.16}$ | $24.35_{\pm1.07}$ | $67.74_{\pm0.70}$ | $80.12_{\pm0.32}$ |
| | | $\Delta$ (%) | -16.80 | -38.22 | +4.80 | -28.52 | -13.93 | -7.68 |
| | $\text{Split-LoRA}_{(4\times8)}$ | Before | $50.68_{\pm1.06}$ | $93.08_{\pm0.41}$ | $77.12_{\pm0.76}$ | $48.65_{\pm1.18}$ | $79.30_{\pm0.91}$ | $86.05_{\pm0.44}$ |
| | | After | $43.56_{\pm0.84}$ | $62.15_{\pm0.26}$ | $81.82_{\pm0.20}$ | $23.79_{\pm0.52}$ | $68.74_{\pm0.96}$ | $81.53_{\pm1.07}$ |
| | | $\Delta$ (%) | -7.12 | -30.93 | +4.70 | -24.86 | -10.56 | -4.52 |
| | $\text{FlyLoRA}_{(k=8)}$ | Before | $53.68_{\pm0.47}$ | $95.55_{\pm0.18}$ | $80.82_{\pm0.56}$ | $54.34_{\pm2.13}$ | $82.85_{\pm0.52}$ | $89.63_{\pm0.55}$ |
| | | After | $61.35_{\pm0.47}$ | $72.94_{\pm0.21}$ | $86.34_{\pm0.36}$ | $34.47_{\pm0.44}$ | $75.97_{\pm0.70}$ | $87.64_{\pm1.04}$ |
| | | $\Delta$ (%) | **+7.67** | **-22.61** | **+5.52** | **-19.87** | **-6.88** | **-1.99** |

Table 12: **Multi-task Performance Comparison with More Advanced Merging Techniques using Qwen-2.5-7B.** We evaluate different methods across MMLU, ScienceQA, GSM8K (accuracy), and HumanEval (Pass@k) benchmarks. The table shows the relative performance drop ($\Delta\%$) before and after merging. The best results are highlighted in **bold**.

| Method | MMLU | ScienceQA | GSM8K | HumanEval |
|---|---|---|---|---|
| FlyLoRA | +6.55 | -23.77 | +4.80 | -21.23 |
| KnOTS | +10.76 | -26.85 | +4.68 | -23.37 |
| L-LoRA | +4.51 | -22.48 | +4.74 | -20.85 |
| KnOTS+FlyLoRA | **+11.47** | -23.41 | **+5.25** | -20.69 |
| L-LoRA+FlyLoRA | +7.65 | **-21.42** | +5.02 | **-19.85** |

These findings collectively demonstrate that biologically inspired top-$k$ activation optimally balances efficiency and task-specific feature selection.

## B.7 Additional Ablation Studies on Matrix A initialization Schemes

We further compare three methods for generating the sparse projection $A$ with $\frac{p}{r}$ sparsity:

1. Gaussian (our default): Each non-zero entry is drawn from $\mathcal{N}(0, \frac{1}{r^2})$.

2. Rademacher (non-Gaussian): Each non-zero entry is $\pm\frac{1}{r}$ with equal probability.

3. FJLT [3] (structured projection): $A = PHD$, where $D$ is a random diagonal matrix with independent Rademacher variables on its diagonal, $H$ is a normalized Hadamard matrix, and $P$ enforces the $\frac{p}{r}$ sparsity.

4. Two-Phase (briefly-learned): The non-zero entries of $A$ are trainable for 5% of total steps as warm-up, then frozen for the remainder.

The results, shown in Table 15, indicate that almost all variants perform similarly. Non-Gaussian, structured, or briefly-learned initializations have little impact, except that the briefly-learned scheme

Table 13: **Performance Comparison of Different Load-Balancing Strategies.** Evaluation on MMLU benchmark using Llama-3.1-8B.

| Load-Balancing Strategy | Accuracy (%) |
|---|---|
| Loss-Free | **40.88**$_{\pm\textbf{1.61}}$ |
| Loss-Controlled | 40.59$_{\pm 0.51}$ |
| No Load-Balancing | 37.56$_{\pm 2.87}$ |

Table 14: **Performance Comparison of Different K-Selection Strategies.** Evaluation on MMLU benchmark using Llama-3.1-8B.

| K-Selection Strategy | Accuracy (%) |
|---|---|
| top-$k$ | **40.88**$_{\pm\textbf{1.61}}$ |
| random-$k$ | 40.02$_{\pm 0.26}$ |
| full activation | 39.40$_{\pm 1.14}$ |

shows a noticeable drop after merging. This demonstrates that FlyLoRA is robust to the choice of initialization scheme for matrix $A$, and that learning may break the approximate orthogonality of the random matrix, making it unsuitable.

### B.8 Analysis of the Performance Gap Between Merged and Non-merged Scenarios

In Tables 2, 10, and 11, we can see that ScienceQA usually demonstrates a large performance drop after merging for all LoRA variants. Intuitively, the four tasks—general knowledge understanding (MMLU), scientific question answering (ScienceQA), mathematical reasoning (GSM8K), and code generation (HumanEval)—represent significantly different distributions, so merging their adapters is prone to substantial conflicts.

Empirically, following [73], we use centered kernel alignment (CKA) [33] to quantify the alignment between the output representations of each single-task adapter and the merged adapter. A higher CKA indicates better output alignment, which is likely the inherent reason, and therefore, results in a smaller accuracy drop after merging. Table 16 reports both CKA and accuracy drop ($\Delta$) on Llama-3.1-8B. Since there is no apparent difference between LoRA$_{(r=8)}$, LoRA$_{(r=32)}$, and Split-LoRA$_{(4\times 8)}$ in model merging, we use LoRA$_{(r=8)}$ as a representative to compare with FlyLoRA$_{(k=8)}$. We observe that tasks with lower CKA (especially ScienceQA and GSM8K) suffer the largest accuracy drops. FlyLoRA consistently yields higher CKA than LoRA, which aligns with its consistently smaller $\Delta$. This micro-level analysis corroborates why FlyLoRA outperforms LoRA in heterogeneous-task merging.

## C Detailed Experimental Setting

### C.1 Datasets

To comprehensively evaluate the effectiveness of our proposed method across diverse domains and task types, we conduct extensive experiments on five carefully selected benchmarks. These datasets span critical capabilities including general knowledge reasoning, scientific understanding, mathematical problem solving, and code generation. Table 17 summarizes the key characteristics of each dataset, while detailed descriptions are provided below:

- **MMLU** [25] serves as a comprehensive benchmark for evaluating broad knowledge understanding and reasoning capabilities. It comprises multiple-choice questions spanning 57 distinct academic subjects, ranging from elementary mathematics and US history to computer science and professional law. The diversity of domains makes it particularly suitable for assessing model generalization across different knowledge types.

- **ScienceQA** [48] provides a multimodal framework for science question answering, with content derived from elementary and high school curricula aligned with California Common Core Content Standards. The questions originate from IXL Learning's expert-curated educational resources. Following the established practice in [37], we utilize only the textual components to focus on linguistic understanding.

Table 15: **Performance Comparison of Different A Initialization Strategies.** Single-task and multi-task accuracy comparison on MMLU using Llama-3.1-8B.

| A Initialization Strategy | Before | $\Delta$after merging |
|---|---|---|
| Gaussian | $40.88_{\pm 1.61}$ | -2.02 |
| Rademacher | $40.42_{\pm 0.23}$ | -2.35 |
| FJLT | $40.57_{\pm 1.34}$ | -2.50 |
| Two-Phase | $40.76_{\pm 1.04}$ | -4.86 |

Table 16: **CKA and Corresponding Accuracy Drop ($\Delta$) Between Single-Task Adapter and Merged Model.** Evaluating using Llama-3.1-8B.

| Method | Task | MMLU | ScienceQA | GSM8K | HumanEval |
|---|---|---|---|---|---|
| LoRA$_{(r=8)}$ | CKA | 0.78 | 0.39 | 0.58 | 0.75 |
| | $\Delta$ | -6.48 | -60.34 | -30.15 | -13.04 |
| FlyLoRA$_{(k=8)}$ | CKA | **0.85** | **0.53** | **0.71** | **0.84** |
| | $\Delta$ | **-2.02** | **-43.05** | **-21.81** | **-4.27** |

- **GSM8K** [12] offers 8,500 high-quality grade school mathematics word problems that demand multi-step arithmetic reasoning. Each problem is accompanied by a detailed, step-by-step solution, making it ideal for evaluating logical reasoning and procedural accuracy in mathematical contexts.

- **CodeAlpaca-20k** [7] contains 20,022 synthetically generated instruction-response pairs specifically designed for code-related tasks. This dataset facilitates effective instruction tuning for programming applications by providing diverse coding prompts paired with corresponding solutions.

- **HumanEval** [9] consists of 164 hand-crafted Python programming problems developed to assess functional correctness in code generation. Crucially, these problems were manually created to prevent data contamination, ensuring they do not appear in the training corpora of existing code generation models.

Table 17: **Details of MMLU, ScienceQA, GSM8K, CodeAlpaca and HumanEval Datasets.** We list the number of training and testing samples and task types for the following datasets used in our experiments.

| Dataset | Training Samples | Testing Samples | Task Types |
|---|---|---|---|
| MMLU [25] | 99,842 | 14,042 | Multiple Choice |
| ScienceQA [48] | 12,726 | 4,241 | Multiple Choice |
| GSM8K [12] | 7,473 | 1,319 | Math Problems |
| CodeAlpaca-20k [7] | 20,022 | — | Code Instruction |
| HumanEval [9] | — | 164 | Code Generation |

## C.2 Training Configuration

This section elaborates on the experimental setup and hyperparameter configurations employed throughout our study. Table 18 documents the shared training parameters applied consistently across all backbone models and datasets. To address the specific requirements of different tasks and model architectures, we additionally provide dataset-specific and model-specific configurations in Table 19, including learning rate schedules, batch size adjustments, and task-specific optimization strategies.

## C.3 Split-LoRA

We implement Split-LoRA as a representative MoE-based LoRA method, following the general framework described in Section 2.2. In our experiments, we incorporate a sigmoid activation function in the router to normalize expert selection scores. Thus, the gating function operates as $G(x) = \text{sigmoid}(\text{top-}k(W_g x))$, which ensures differentiable routing while maintaining the sparsity of expert activation. This configuration allows Split-LoRA to serve as a representative baseline for evaluating the effectiveness of MoE structures in LoRA.

Table 18: **General Training Hyperparameters for FlyLoRA.** Shared configuration across all experiments, including rank settings, optimizer details, and architectural choices.

| Parameter | Value |
|---|---|
| Total rank ($r$) | 32 |
| Scaling factor ($\alpha$) | 64 |
| Activated rank | 8 |
| Target modules | {q,k,v,o,gate,down,up}_proj |
| Optimizer | AdamW |
| Warmup ratio | 0.01 |
| Gradient accumulated batch | 128 |
| Dropout rate | 0.00 |

Table 19: **Dataset-Specific and Model-Specific Training Configurations for FlyLoRA.** Task-optimized settings for Llama-3.1-8B and Qwen-2.5-7B across four benchmarks, showing variations in epoch counts, learning rates, and sequence lengths based on dataset characteristics and model requirements.

| Model | Parameter | MMLU | ScienceQA | GSM8K | CodeAlpaca |
|---|---|---|---|---|---|
| Llama-3.1-8B | Epochs | 1 | 20 | 1 | 2 |
| | Learning rate | $3 \times 10^{-4}$ | $3 \times 10^{-4}$ | $3 \times 10^{-4}$ | $3 \times 10^{-4}$ |
| | Max sequence length | 128 | 256 | 512 | 512 |
| | micro batch size | 8 | 8 | 8 | 8 |
| Qwen-2.5-7B | Epochs | 1 | 20 | 1 | 2 |
| | Learning rate | $3 \times 10^{-4}$ | $3 \times 10^{-4}$ | $3 \times 10^{-4}$ | $6 \times 10^{-4}$ |
| | Max sequence length | 128 | 256 | 512 | 512 |
| | micro batch size | 8 | 8 | 8 | 8 |

## C.4 Environments

Most experiments were conducted on a Linux server running Ubuntu 20.04.4 LTS, equipped with an Intel(R) Xeon(R) Platinum 8358P CPU at 2.60GHz and 8 NVIDIA GeForce RTX 3090 GPUs, using CUDA version 11.7. Experiments with Qwen-2.5-14B were conducted on a machine with 8 NVIDIA A100 GPUs.

## D    Limitations and Future Work

In FlyLoRA, matrix $A$ is randomly initialized and frozen during training, but there may still be room for improvement. Recent neuroscience studies [13] suggest that $A$ need not be entirely frozen and random, indicating potential for more bio-inspired mechanisms to enhance task decoupling through an adaptable version of $A$. Moreover, recent works suggest that component-wise interpretability [98] and spectral modulation [74] could inspire adaptive or frequency-aware modifications of $A$ in FlyLoRA to improve efficiency, robustness, and task decoupling.

Recently, RL fine-tuning for LLMs has emerged as a promising approach that significantly enhances their reasoning ability [24]. However, stabilizing MoE RL training remains an open question [96], and further exploration will focus on the integration of FlyLoRA with RL training [61] and potentially extending it to offline policy optimization [50–53, 60, 68]. Additionally, integrating active data selection methods could be a promising direction to further improve data efficiency [62, 77, 79, 99].

## E    Broader Impact

Our proposed FlyLoRA resolves the trade-off between parameter interference and efficiency in MoE-based LoRA approaches. Additionally, this efficient decoupling mechanism, which is inspired by fly olfactory circuits, can be applied across various domains, helping researchers and developers leverage more powerful LoRA fine-tuning strategies. On the other hand, FlyLoRA could potentially be misused to fine-tune LLMs that exhibit biases or generate harmful content. We recommend implementing model access controls and bias-monitoring frameworks when deploying this technique.

