# OpenReview forum: "FlyLoRA: Boosting Task Decoupling and Parameter Efficiency via Implicit Rank-Wise Mixture-of-Experts"
_NeurIPS.cc/2025/Conference — NeurIPS 2025 poster_

### Official Review · Reviewer_ruPf · 2025-06-29

**Clarity:** 3
**Significance:** 3
**Originality:** 3
**Rating:** 4
**Confidence:** 4

**Summary:**

This paper introduces FlyLoRA, a novel PEFT) method inspired by the fly olfactory circuit. FlyLoRA addresses the limitations of existing LoRA methods, specifically parameter interference and computational inefficiency, through implicit rank-wise expert activation and frozen sparse random down-projections. The method theoretically decouples intra- and inter-task interactions, leading to enhanced decorrelation in both single-task instruction tuning and multi-task model fusion. Experimental results demonstrate consistent performance improvements over existing PEFT methods and significantly reduced performance degradation in multi-task merging scenarios.

**Questions:**

Please refer to weaknesses.

**Ethical Concerns:**

["NO or VERY MINOR ethics concerns only"]

**Final Justification:**

The paper is quite solid and interesting, and below are the final justifications.
1. The paper introduces FlyLoRA, which is a compelling and innovative design.
2. The paper explains why there is such a significant difference in results between merged and non-merged scenarios and addresses my concern.
3. The evaluation results are comprehensive enough, which include more challenging and common benchmarks such as GPQA, LiveCodeBench, and advanced Math reasoning datasets.

**Limitations:**

yes

**Paper Formatting Concerns:**

No formatting issues.

**Quality:**

3

**Strengths And Weaknesses:**

Strengths:
1. FlyLoRA proposes a unique approach to PEFT by incorporating implicit rank-wise expert activation and frozen sparse random projections, inspired by biological neural structures, which is a compelling and innovative design.
2. By eliminating the need for explicit router parameters and leveraging inherent sparsity, FlyLoRA offers improved computational efficiency compared to other MoE-based LoRA variants.

Weaknesses:
1. The paper does not sufficiently explain why there is such a significant difference in results between merged and non-merged scenarios. A deeper analysis of this discrepancy would enhance the understanding of FlyLoRA's behavior in diverse contexts.
2. The evaluation results, while promising, are not comprehensive enough. The paper would benefit from testing on more challenging and common benchmarks such as GPQA, LiveCodeBench, and advanced Math reasoning datasets, to thoroughly assess FlyLoRA's robustness and scalability.

---

> ### Author Rebuttal · Authors · 2025-07-31
>
> We appreciate the reviewer's valuable feedback and hope our responses can address the raised concerns.
>
> **W1: Analysis of the performance gap between merged and non-merged scenarios**
>
> Intuitively, the four tasks—general knowledge understanding (MMLU), scientific question answering (ScienceQA), mathematical reasoning (GSM8K), and code generation (HumanEval)—represent significantly different distributions, so merging their adapters is prone to substantial conflicts.
>
> Empirically, following [1], we use centered kernel alignment (CKA) [2] to quantify the alignment between the output representations of each single-task adapter and the merged adapter. A higher CKA indicates better output alignment, which is likely the inherent reason, and therefore, results in a smaller accuracy drop after merging. Supplementary Table 1 reports both CKA and accuracy drop ($\Delta$) on Llama-3.1-8B. Since there's no apparent difference between LoRA$(r=8)$, LoRA$(r=32)$ and Split-LoRA$(4 \times 8)$ in model merging, we use LoRA$(r=8)$ as a representative to compare with FlyLoRA$(k=8)$. We observe that tasks with lower CKA (especially ScienceQA and GSM8K) suffer the largest accuracy drops. FlyLoRA consistently yields higher CKA than LoRA, which aligns with its consistently smaller $\Delta$. This micro-level analysis corroborates why FlyLoRA outperforms LoRA in heterogeneous-task merging. All of this new analysis will be included in the final version.
>
> **Supplementary Table 1: CKA and corresponding accuracy drop ($\Delta$) between single-task adapter and merged model.**
>
> |Method|Task|MMLU|ScienceQA|GSM8K|HumanEval|
> |:-:|:-:|:-:|:-:|:-:|:-:|
> |LoRA$_{(r=8)}$|CKA|0.78|0.39|0.58|0.75|
> ||$\Delta$|-6.48|-60.34|-30.15|-13.04|
> |FlyLoRA$_{(k=8)}$|CKA|**0.85**|**0.53**|**0.71**|**0.84**|
> ||$\Delta$|**-2.02**|**-43.05**|**-21.81**|**-4.27**|
>
> **W2: Evaluating on more challenging benchmarks**
>
> As suggested by the reviewer, we further evaluated single- and multi-task performance on LiveCodeBench (using CodeAlpaca checkpoints), GPQA (using MMLU checkpoints), and MATH (for advanced math reasoning, using GSM8K checkpoints) with Qwen-2.5-7B. The results in Supplementary Tables 2 and 3 indicate that FlyLoRA remains robust even on more challenging test sets.
>
> **Supplementary Table 2: Single-task accuracy comparison (Pass@1 for LiveCodeBench) using Qwen-2.5-7B.**
>
> |Method|Param(%)|GPQA|MATH|LiveCodeBench|
> |:-:|:-:|:-:|:-:|:-:|
> |LoRA$_{(r=8)}$|0.26|34.9±0.6|45.7±0.4|12.4±0.1|
> |LoRA$_{(r=32)}$|1.05|36.8±0.2|47.7±0.1|14.8±0.2|
> |LoRA-Split$_{(4\times 8)}$|0.33|36.2±0.3|46.9±0.3|14.5±0.1|
> |**FlyLoRA$_{(k=8)}$**|**0.13**|**37.4±0.3**|**48.1±0.4**|**15.2±0.1**|
>
> **Supplementary Table 3: Multi-task accuracy comparison (Pass@1 for LiveCodeBench) using Qwen-2.5-7B. We report the accuracy drop relative to single-task performance using weight averaging technique.**
>
> |Method|GPQA|MATH|LiveCodeBench|
> |:-:|:-:|:-:|:-:|
> |LoRA$_{(r=8)}$|-0.5|-4.1|-3.8|
> |LoRA$_{(r=32)}$|-1.6|-4.4|-3.4|
> |LoRA-Split$_{(4\times 8)}$|+0.4|-5.2|-3.5|
> |**FlyLoRA$_{(k=8)}$**|**+1.3**|**-2.4**|**-1.8**|
>
> **References**
>
> [1] Model merging with SVD to tie the Knots. ICLR2025.
>
> [2] Similarity of Neural Network Representations Revisited. ICML 2019.

---

> > ### Comment · Reviewer_ruPf · 2025-08-04
> >
> > Thanks for the clarification. The reviewer's comments address my concern, and I will raise the final rating.

---

> ### Author Response · Authors · 2025-08-04
> **Author Response to Reviewer ruPf's Comment**
>
> Thanks for the positive feedback. We are delighted that our response addressed your concerns. We will incorporate these discussions and relate new results into our final version.

---

### Official Review · Reviewer_oMGp · 2025-07-02

**Clarity:** 3
**Significance:** 3
**Originality:** 4
**Rating:** 5
**Confidence:** 4

**Summary:**

This paper proposes FlyLoRA, a novel LoRA variant inspired by the fly olfactory circuit, which improves parameter efficiency and task decoupling in both single-task and multi-task settings. The key innovations include: (1) using a frozen sparse random projection matrix $A$ to replace the conventional trainable down-projection, and (2) performing rank-wise top-k expert activation in the up-projection matrix $B$, eliminating the need for an explicit router. Theoretical analysis shows that the fixed sparse matrix preserves input distances and induces near-orthogonal subspaces, which mitigates parameter interference across tasks. Empirical results across several benchmarks using two backbones show FlyLoRA achieves strong performance with lower memory and compute overhead.

**Questions:**

- Can the authors discuss the sensitivity of the frozen matrix $A$ to different initialization schemes? Could non-Gaussian or structured projections affect performance?
- How does FlyLoRA perform on significantly larger models (> 8B)? Are there bottlenecks in memory or convergence?
- Can more advanced MoE-based LoRA baselines be included for comparison?
- What causes the large performance drop after merging FlyLoRA on ScienceQA & GSM8K tasks?
- Could ST+Trn outperform ST+Frz on some tasks, and if so, what trade-offs would it bring in merging or stability?

**Ethical Concerns:**

["NO or VERY MINOR ethics concerns only"]

**Final Justification:**

* The authors provided strong additional results in the rebuttal, including sensitivity analysis for the frozen projection matrix, evaluations on a 14B model, and comparisons with stronger MoE-based and adaptive baselines.
* The explanation of merging-related performance drops using CKA and the detailed ablation studies between ST+Trn and ST+Frz effectively addressed my earlier concerns.
* While some trade-offs remain inherent to the merging process, the core claims are now better supported both theoretically and empirically.
* Overall, the rebuttal meaningfully strengthened the paper and justified an increase in my score.

**Limitations:**

Yes.

**Paper Formatting Concerns:**

None.

**Quality:**

3

**Strengths And Weaknesses:**

**Strengths**:

* Quality: The paper presents a well-grounded method with both strong theoretical justifications (distance preservation, covariance reduction, orthogonality) and extensive empirical validation.
* Clarity: The paper is well-written with clear motivation, architecture explanation, and figures. The supplementary materials provide thorough theoretical proofs and ablation studies.
* Significance: FlyLoRA addresses a core limitation in existing LoRA and MoE-LoRA methods—parameter interference while improving computational efficiency. The method has broad applicability in fine-tuning large models under resource constraints.
* Originality: The idea of combining sparse random projections with rank-wise MoE and top-k implicit routing, inspired by neuroscience, is novel and elegant. The routing-free design simplifies architecture while retaining strong performance.

**Weaknesses**:

* The initialization of the sparse random projection matrix $A$ uses a fixed Gaussian scheme. It remains unclear how sensitive the method is to this choice. Could other distributions or learned schemes improve or worsen performance?
* Evaluation on larger backbone models is missing. It is unclear how FlyLoRA scales with increasing model sizes beyond 8B.
* Only one MoE-LoRA baseline (Split-LoRA) is included. More recent or stronger adaptive/sparse fine-tuning baselines could strengthen the comparison.
* In Table 2, performance of FlyLoRA + Llama-3.1-8B drops sharply on ScienceQA and GSM8K after merging. A deeper analysis of these drops would be useful.
* The ablation in Table 3 shows marginal difference between ST+Trn and ST+Frz. It's unclear whether ST+Trn might outperform ST+Frz on other benchmarks.

---

> ### Author Rebuttal · Authors · 2025-07-31
>
> We appreciate the reviewer's valuable feedback and hope our responses can address the raised concerns.
>
> **W1&Q1: More sensitivity studies on initialization schemes of matrix $A$**
>
> As the reviewer suggested, we further compare three methods for generating the sparse projection $A$ with $\frac{p}{r}$ sparsity:
>
> - Gaussian (our default): Each non-zero entry is drawn from $\mathcal{N}(0,\frac{1}{r^2})$；
> - Rademacher (non-Gaussian): Each non-zero entry is $±\frac{1}{r}$ with equal probability.
> - FJLT [1] (structured projection): $A=PHD$, where $D$ is a random diagonal matrix with independent Rademacher variables on its diagonal, $H$ is a normalized Hadamard matrix, and $P$ enforces the $\frac{p}{r}$ sparsity.
> - Two-Phase (briefly-learned): The non-zero entries of $A$ are trainable for 5% of the total steps (warm-up), then frozen for the remainder.
>
> The results, shown in Supplementary Table 1, indicate that almost all variants perform similarly. Non-Gaussian, structured, or briefly-learned initializations have little impact, except that the briefly-learned scheme shows a noticeable drop after merging. This demonstrates that FlyLoRA is robust to the choice of initialization scheme for matrix $A$, and that learning may break the approximate orthogonality of the random matrix, making it unsuitable.
>
> **Supplementary Table 1: Single-task and multi-task accuracy comparison on MMLU using Llama-3.1-8B.**
>
> |Method|before|$\Delta$ after merging|
> |:-:|:-:|:-:|
> |Gaussian|40.88±1.61|-2.02|
> |Rademacher|40.42±0.23|-2.35|
> |FJLT|40.57±1.34|-2.50|
> |Two-Phase|40.76±1.04|-4.86|
>
> **W2&Q2: Evaluation of larger models**
>
> We further conducted experiments using the Qwen-2.5-14B model, as shown in Supplementary Tables 2 and 3, following the setting of Tables 1 and 2. The results show that FlyLoRA remains superior in accuracy (for both single-task and multi-task settings) and is more parameter-efficient. We encountered no memory or convergence bottlenecks when training FlyLoRA on the 14B model, which consistently outperforms LoRA.
>
> **Supplementary Table 2: Single-task accuracy comparison (Pass@1 for HumanEval) using Qwen-2.5-14B.**
>
> |Method|Param(%)|MMLU|ScienceQA|GSM8K|HumanEval|
> |:-:|:-:|:-:|:-:|:-:|:-:|
> |LoRA$_{(r=8)}$|0.23|56.74±0.56|95.62±0.18|83.08±0.74|51.69±1.48|
> |LoRA$_{(r=32)}$|0.93|59.35±0.79|97.05±0.22|85.31±0.25|54.80±0.76|
> |LoRA-Split$_{(4\times 8)}$|0.29|58.26±1.13|96.85±0.35|84.88±0.51|54.65±0.59|
> |**FlyLoRA$_{(k=8)}$**|**0.12**|**60.17±1.08**|**97.37±0.32**|**85.96±0.89**|**56.42±1.16**|
>
> **Supplementary Table 3: Multi-task accuracy comparison (Pass@1 for HumanEval) using Qwen-2.5-14B. We report the accuracy drop relative to single-task performance using weight averaging technique.**
>
> |Method|MMLU|ScienceQA|GSM8K|HumanEval|
> |:-:|:-:|:-:|:-:|:-:|
> |LoRA$_{(r=8)}$|-13.75|-25.20|-11.43|-18.60|
> |LoRA$_{(r=32)}$|-8.91|-20.45|-7.62|-16.34|
> |LoRA-Split$_{(4\times 8)}$|-7.48|-21.97|-6.05|-14.87|
> |**FlyLoRA$_{(k=8)}$**|**-4.35**|**-17.89**|**-2.18**|**-11.72**|
>
> **W3&Q3: More baseline into comparison**
>
> We further compare several strong and widely used baselines—AdaLoRA [2] (adaptive), SoRA [3] (sparse), and HydraLoRA [4] (MoE)—using Qwen-2.5-7B, as shown in Supplementary Tables 4 and 5. The results demonstrate that FlyLoRA remains superior in terms of accuracy and efficiency in both single-task learning and multi-task merging scenarios.
>
> **Supplementary Table 4: Single-task accuracy comparison (Pass@1 for HumanEval) using Qwen-2.5-7B.**
>
> |Method|Param(%)|MMLU|ScienceQA|GSM8K|HumanEval|
> |:-:|:-:|:-:|:-:|:-:|:-:|
> |AdaLoRA$_{(r=8)}$|0.26|51.22±0.21|93.48±0.28|77.65±0.14|47.96±1.34|
> |SoRA$_{(r=8)}$|0.19|50.89±0.42|93.25±0.20|78.46±0.82|47.83±0.94|
> |HydraLoRA$_{(r=8, A=1, B=3)}$|0.52|53.05±0.16|94.69±0.34|79.31±0.49|52.98±1.57|
> |**FlyLoRA$_{(k=8)}$**|**0.13**|**53.68±0.47**|**95.55±0.18**|**80.82±0.56**|**54.34±2.13**|
>
> **Supplementary Table 5: Multi-task accuracy comparison (Pass@1 for HumanEval) using Qwen-2.5-7B. We report the accuracy drop relative to single-task performance using weight averaging technique.**
>
> |Method|MMLU|ScienceQA|GSM8K|HumanEval|
> |:-:|:-:|:-:|:-:|:-:|
> |AdaLoRA$_{(r=8)}$|-10.90|-33.15|+4.52|-25.46|
> |SoRA$_{(r=8)}$|-9.45|-34.81|+4.05|-26.50|
> |HydraLoRA$_{(r=8, A=1, B=3)}$|-6.22|-34.67|+4.21|-24.94|
> |**FlyLoRA$_{(k=8)}$**|**+6.55**|**-23.77**|**+4.80**|**-21.23**|
>
> **W4&Q4: Analysis in performance drop between merge and non-merge settings**
>
> The four tasks—general knowledge understanding (MMLU), scientific question answering (ScienceQA), mathematical reasoning (GSM8K), and code generation (HumanEval)—represent significantly different distributions. As a result, merging their adapters is prone to major conflicts that can cause performance drops.
>
> Empirically, following [5], we use centered kernel alignment (CKA) [6] to quantify the alignment between the output representations of each single-task adapter and the merged adapter. A higher CKA indicates better output alignment, which is likely the inherent reason, and therefore, results in a smaller accuracy drop after merging. Supplementary Table 1 reports both CKA and accuracy drop ($\Delta$) on Llama-3.1-8B. Since there's no apparent difference between LoRA$(r=8)$, LoRA$(r=32)$ and Split-LoRA$(4 \times 8)$ in model merging, we use LoRA$(r=8)$ as a representative to compare with FlyLoRA$(k=8)$. We observe that tasks with lower CKA (especially ScienceQA and GSM8K) suffer the largest accuracy drops. This micro-level analysis helps explain the substantial performance decreases observed on ScienceQA and GSM8K. It is important to note that such degradation is an inherent phenomenon of model merging, not a weakness of FlyLoRA. In fact, FlyLoRA still outperforms its LoRA variants (Tables 1 and 2), even in cases where accuracy falls sharply.
>
> **Supplementary Table 6: CKA and corresponding accuracy drop ($\Delta$) between single-task adapter and merged model.**
>
> |Method|Task|MMLU|ScienceQA|GSM8K|HumanEval|
> |:-:|:-:|:-:|:-:|:-:|:-:|
> |LoRA$_{(r=8)}$|CKA|0.78|0.39|0.58|0.75|
> ||$\Delta$|-6.48|-60.34|-30.15|-13.04|
> |FlyLoRA$_{(k=8)}$|CKA|0.85|0.53|0.71|0.84|
> ||$\Delta$|-2.02|-43.05|-21.81|-4.27|
>
> **W5&Q5: More ablation studies between ST+Trn and ST+Frz**
>
> We further conducted ablation studies on the GSM8K, ScienceQA, and HumanEval datasets, with results presented in Supplementary Table 7. The results show that ST+Trn does not outperform ST+Frz, with almost no difference in accuracy. Previous papers have already demonstrated that, in LoRA, updating matrix $A$ is not necessary [7,8]. Our results further validate this finding in the MoE-based LoRA setting. Thus, ST+Trn offers no clear advantage in this context, while ST+Frz can help reduce both interference during merging and the memory footprint of activation values [7].
>
> **Supplementary Table 7: Ablation study of FlyLoRA variants using Llama-3.1-8B.**
>
> |Method|ScienceQA|GSM8K|HumanEval|
> |:-:|:-:|:-:|:-:|
> |ST+Trn|94.10±0.47|58.68±0.52|36.81±1.36|
> |ST+Frz|94.15±0.36|58.76±0.74|36.88±1.91|
>
> **References**
>
> [1] The fast Johnson-Lindenstrauss transform is even faster. ICML2023.
>
> [2] AdaLoRA: Adaptive Budget Allocation for Parameter-Efficient Fine-Tuning. ICLR2023.
>
> [3] Sparse Low-rank Adaptation of Pre-trained Language Models. EMNLP2023.
>
> [4] HydraLoRA: An Asymmetric LoRA Architecture for Efficient Fine-Tuning. Neurips2024.
>
> [5] Model merging with SVD to tie the Knots. ICLR2025.
>
> [6] Similarity of Neural Network Representations Revisited. ICML 2019.
>
> [7] LoRA-FA: Memory-efficient Low-rank Adaptation for Large Language Models Fine-tuning. Arxiv2023.
>
> [8] Asymmetry in Low-Rank Adapters of Foundation Models. ICML2024.

---

> > ### Comment · Reviewer_oMGp · 2025-08-05
> > **Reply to Authors**
> >
> > Thank you for the detailed and thoughtful rebuttal. The additional experiments on initialization sensitivity, larger model scaling, and comparisons with stronger baselines significantly strengthened the submission. I also appreciated the in-depth analysis of merging-related performance drops and the extended ablation studies. These clarifications addressed my key concerns, and I have updated my score accordingly.

---

> > > ### Author Response · Authors · 2025-08-06
> > > **Author Response to Reviewer oMGp's Comment**
> > >
> > > We appreciate the encouraging remarks and are pleased that our reply has resolved your concern. These additional discussions and experiments will be integrated into the final manuscript.

---

### Official Review · Reviewer_vPRz · 2025-07-02

**Clarity:** 4
**Significance:** 4
**Originality:** 4
**Rating:** 5
**Confidence:** 5

**Summary:**

The paper proposes FlyLoRA, a parameter-efficient fine-tuning method inspired by the fly olfactory circuit. It uses a frozen sparse projection for routing and rank-wise expert activation to reduce interference. This design improves both intra-task and inter-task decoupling, enabling efficient single-task learning and robust multi-task model merging. FlyLoRA outperforms standard and MoE-based LoRA methods across multiple domains with fewer trainable parameters.

**Questions:**

1. Corollary 3.4: How can the additive norm preservation property benefit model merging? The orthogonality part is intuitive.
2. Why can we apply Theorem A.1 to Theorem A.2 where in the statement of A.1, $\mathbf{A}$ is a random matrix sampled iid from standard normal, but in A.2 the matrix $\mathbf{A}$ has exactly $p$ non-zero entries as how you construct the matrix in FlyLoRA.
3. Can you explain why Theorem 3.1 extends to Top-k activation from Equation (2) in the Appendix?
4. For A.3, it’s better to move the zero-mean gradient assumption (line 36) above and make it explicit in the theorem statement.
5. In Table 2, the current baselines may be too weak because there are many papers focusing on reducing interference of LoRA for merging but split-LoRA is designed to improve single task PEFT performance. For example, you can try to compare with [1,2].

[1] Stoica, George, et al. "Model merging with SVD to tie the Knots." The Thirteenth International Conference on Learning Representations.

[2] Tang, Anke, et al. "Parameter-Efficient Multi-Task Model Fusion with Partial Linearization." The Twelfth International Conference on Learning Representations. 2023.

**Ethical Concerns:**

["NO or VERY MINOR ethics concerns only"]

**Final Justification:**

Although this paper still has some glitches for the mathematical rigor that cannot exactly explain the FlyLoRA properties (thus not a higher score), overall this is a high-quality paper, and the natural orthogonality with the random matrices is a very clever idea. Good job!

**Limitations:**

Yes, in Appendix E.

**Paper Formatting Concerns:**

NA.

**Quality:**

3

**Strengths And Weaknesses:**

## Strength
**Quality**: The paper is overall technically sound. Most empirical claims are well supported by experiments. The methods used are well-motivated by both biological inspirations, classic random matrix theories and very practical concerns for the trade-off between MoE-based LoRA performance and efficiency. The authors mentioned the weaknesses in the appendix and addressed it properly.

**Clarity**: The submission is clearly written and well-organized and based on the writing it should be easy to reproduce based on the figure visualization and equation 11. The authors are able to guide readers through the motivation to propose FlyLoRA step by step from section 2 to 3.

**Significance**: The results will be very impactful: first the FlyLoRA shows better generalization performance with lower parameters; second FlyLoRA shows better performance also on PEFT model merging, and both first and second topic are very widely-used techniques; third all theoretical analysis can be proved by empirical advantages.

**Originality**: To my best knowledge the formulation of FlyLoRA is new, although some parts of the theoretical analysis framework are similar to Johnson–Lindenstrauss analysis. This is not a weakness though, as I appreciate authors who show how to apply these frameworks to the setup of FlyLoRA. Theorem 3.2 is also insightful to explain the success of TopK sparsity in other applications.

## Weaknesses
**Quality**: I have a few important clarification questions for the theoretical section. Please see Questions for details. Besides, the model merging part would be strengthened by more comprehensive LoRA merging baselines.

---

> ### Author Rebuttal · Authors · 2025-07-31
>
> We appreciate the reviewer's valuable feedback and hope our responses can address the raised concerns.
>
> **Q1: How can additive norm preservation property benefit for model merging**
>
> "Additive Norm Preservation" is a direct consequence of the "Pairwise Orthogonality" property described in Corollary 3.4. Both "Pairwise Orthogonality" and "Additive Norm Preservation" reflect the orthogonality property of FlyLoRA during model merging. According to the widely used geometric intuition (an assumption) that orthogonality benefits model merging [1–3], FlyLoRA's design aligns with this principle.
>
> **Q2: How can i.i.d sampling assumption applied to constraint on number of elements of each entry?**
>
> We will modify the description in Theorem A.2 to: "Given the matrix $A\in\mathbb{R}^{r\times n}$ with each entry i.i.d. from $\mathcal{N}(0, \frac{1}{r^2})$ with probability $\frac{r}{p}$, and set to $0$ otherwise," to make the proof more rigorous. Since this construction does not differ significantly from our original construction for $A$, we use this more analytically tractable form as a surrogate to study the distance-preserving property. We also compare the empirical performance of these two construction methods using Llama-3.1-8B in Supplementary Table 1 to support this approximation, with the difference in accuracy being negligible (all differences are within 0.2%). We will clarify this point in the final version.
>
> **Supplementary Table 1: Accuracy comparison of FlyLoRA using i.i.d versus Fixed-$p$ sparsity patterns.**
>
> |Method|MMLU|$\Delta$ after merging|
> |:-:|:-:|:-:|
> |gaussian i.i.d. p|40.79±1.46|-2.13|
> |gaussian fixed p|40.88±1.61|-2.02|
>
> **Q3: How does Theorem 3.1 extend to Top-k activation?**
>
> We apologize for the confusion. The goal of an MoE router is to select $k$ out of $r$ experts that best approximate the effect of all $r$ experts. In FlyLoRA, according to Theorem 3.1 and Appendix A.2, we assume that the value of $a_i x$ for $i = 1, 2, \ldots, r$ is sufficient to serve as a surrogate for identifying the most informative experts. The top-$k$ operation, therefore, selects experts according to the magnitude of $a_i x$. We will clarify this assumption and its logical connection in the final version.
>
> **Q4: Placement of assumption**
>
> Thanks for the suggestion. We will revise our text as recommended by the reviewer.
>
> **Q5: Comparison with other merging technique**
>
> We include the comparison in Supplementary Table 2. KnOTS [4] and L-LoRA [3] are both built upon LoRA$_{(r=32)}$. The results suggest that they achieve comparable performance to FlyLoRA, with each method excelling on different datasets. It is noteworthy that FlyLoRA is not a competitor to these methods; rather, they can be used in a plug-and-play manner with FlyLoRA to further improve performance after merging.
>
> **Supplementary Table 2: Multi-task accuracy comparison (Pass@1 for HumanEval) using Qwen-2.5-7B. We report the accuracy drop relative to single-task performance.**
>
> |Method|MMLU|ScienceQA|GSM8K|HumanEval|
> |:-:|:-:|:-:|:-:|:-:|
> |FlyLoRA|+6.55|-23.77|+4.80|-21.23|
> |KnOTS|+10.76|-26.85|+4.68|-23.37|
> |L-LoRA|+4.51|-22.48|+4.74|-20.85|
> |KnOTS+FlyLoRA|**+11.47**|-23.41|**+5.25**|-20.69|
> |L-LoRA+FlyLoRA|+7.65|**-21.42**|+5.02|**-19.85**|
>
> **References**
>
> [1] Editing Models with Task Arithmetic. ICLR2023.
>
> [2] Task Arithmetic in the Tangent Space: Improved Editing of Pre-Trained Models. Neurips2023.
>
> [3] Parameter-Efficient Multi-Task Model Fusion with Partial Linearization. ICLR2023.
>
> [4] Model merging with SVD to tie the Knots. ICLR2025.

---

> ### Comment · Reviewer_vPRz · 2025-08-01
>
> I appreciate the author's reply for clarification of some assumptions. The experiments part is now much stronger with additional experiments on KnoTs and L-LoRA! Glad to see this! I have a few more questions and suggestions:
> > "Additive Norm Preservation" is a direct consequence of the "Pairwise Orthogonality"
>
> Although from [1-3]  FlyLoRA's design aligns with this principle of orthogonality ([5,6] also theoretcially proves this), it's still a bit confusing to show the additive norm preservation property because my understanding is that it's not directly related to model merging but instead just some nice property of your FlyLoRA formulation? If it's correct, I would recommend authors to 1) cite the orthogonality intuition papers accordingly 2) clarify the implication of the second norm preservation property.
>
> [5] Efficient Model Editing with Task Vector Bases: A Theoretical Framework and Scalable Approach.
>
> [6] When is Task Vector Provably Effective for Model Editing? A Generalization Analysis of Nonlinear Transformers
>
> > What's the point of Theorem 3.1?
>
> Although 3.1 is a nice property due to your random matrix construction, somehow I was thinking about your paper after the reviewing period and came up with this question: does MoE router have to preserve distance? Why is "no information loss" after projection important as a router? Why can't we just use any gating network that takes $x$ as an input and learn it like the standard MoE which I believe doesn't nessarily have this property and learn very well?

---

> ### Author Response · Authors · 2025-08-03
> **Author Response to Reviewer vPRz's Comment**
>
> We are pleased that our response has addressed the reviewer's concerns, and we appreciate the reviewer's insightful suggestions, which have helped improve our work!
>
> > "Additive Norm Preservation" is a direct consequence of the "Pairwise Orthogonality"
>
> To be more precise, we will modify the "Additive Norm Preservation" property into "Orthogonality's Outcome in Merging". Our original motivation for presenting "Orthogonality's Outcome in Merging" was the concern that "Pairwise Orthogonality" is only a property of LoRA components, which might not be directly linked to the computational steps of model merging. So, we added the mathematical outcome when "Pairwise Orthogonality" is directly applied to the model-merging formula. These two properties are both related to model merging.
>
> As suggested by the reviewer, we will cite the relevant orthogonality-related papers and present the geometric intuition more clearly in the revised version.
>
> > What's the point of Theorem 3.1?
>
> The reviewer is correct that traditional MoE routers neither have nor require the distance-preserving property, as these routers are **trainable** and can adjust their weights to optimize routing. In contrast, the frozen matrix $A$ in FlyLoRA is **not trainable** and therefore cannot adapt in the same way.
>
> Because of the distance-preserving property of the frozen $A$, two semantically similar inputs $x$ and $y$ are projected to similar low-dimensional representations, $Ax$ and $Ay$. Points that are close in the input space remain close after projection, while dissimilar points remain far apart. Consequently, the top-$k$ operation routes similar inputs to a similar set of experts, and dissimilar inputs to different experts. This helps mitigate expert representation homogenization in MoE models, enabling each expert to focus on specialized knowledge, reducing internal conflicts, and improving sample efficiency (as supported by previous studies, e.g., [7]). This is the main point of Theorem 3.1 and the reason why the frozen $A$ is effective.
>
> We will revise Section 3.2 according to the above discussion. The phrase "no information loss" will be removed in the revised version.
>
> [7] Patch-level Routing in Mixture-of-Experts is Provably Sample-efficient for Convolutional Neural Networks. ICML2023.

---

### Official Review · Reviewer_gKhQ · 2025-07-03

**Clarity:** 4
**Significance:** 3
**Originality:** 4
**Rating:** 6
**Confidence:** 5

**Summary:**

This paper begins with the elegant observation that increasing number of experts while fixing total rank monotonically improves LoRA performance. However, existing routing mechanisms increase in parameter count linearly with number of experts, resulting in worse efficiency. To enable large numbers of low-rank experts as an efficient strategy, the authors introduce FlyLoRA, which removes the expensive routing mechanism in MoE-LoRA methods in favour of randomly initialised sparse (and provably near-orthogonal) random A projection, only learning the decoder B. This achieves empirically high performance on a variety of benchmarks against MoE- and standard LoRA, and has less degradation when merged.

**Questions:**

- Previous PEFT methods have used orthogonality as a design decision with success even without MoE, e.g. [OFT](https://arxiv.org/abs/2311.06243), [ReFT](https://proceedings.neurips.cc/paper_files/paper/2024/hash/75008a0fba53bf13b0bb3b7bff986e0e-Abstract-Conference.html). Why do you think this is the case, and is it related to why FlyLoRA is successful (beyond just the merging setting)? If you enforced orthogonality on the A matrix, would that help performance?

**Ethical Concerns:**

["NO or VERY MINOR ethics concerns only"]

**Final Justification:**

Satisfied by author response.

**Limitations:**

yes

**Quality:**

4

**Strengths And Weaknesses:**

Strengths
- The paper is elegantly characterised, explaining and empirically justifying each of the design decisions in FlyLoRA with experiments and theory. Overall I found the context well-presented and easy to understand.
- The method consistently achieves large performance gains over baselines and also has less degradation when merged, while being more parameter-efficient than LoRA and MoE variants.
- The ablations in table 3 and figure 4 show that the theoretical claims made by the authors seem to hold up empirically, particularly (which I found surprising) frozen near-orthogonal A seems actively helpful beyond the multi-task merging setting.
- The biological justification is quite interesting!

Weaknesses
- None apparent.

Typos
- line 206: updateble

---

> ### Author Rebuttal · Authors · 2025-07-31
>
> We appreciate the reviewer's valuable feedback and hope our responses can address the raised concerns.
>
> **Q1: Connection between FlyLoRA and orthogonality design**
>
> Both OFT [1,2] and LoReFT [3] operate on single tasks, but their orthogonal matrix $R$ is **multiplied** with the pre-trained weight matrix $W_0$, which differs from all LoRA variants (including FlyLoRA) that **add** $\Delta W$ to $W_0$. The multiplication scheme **rotates the entire weight space**, and [1,2] demonstrate that this approach better adjusts semantic information compared to changing the magnitude. This is why the scheme succeeds. The additive scheme lacks this property, as $W_0$ cannot be rotated. Thus, although all use an orthogonality design, FlyLoRA succeeds in a different way.
>
> By removing the MoE part and keeping only the random matrix $A$, FlyLoRA reduces to LoRA-FA [4] / Asymmetry LoRA [5], which saves resources but does not improve performance. We also tested enforcing the orthogonality constraint ($A^\top A=I$) on an updatable $A$ (see Supplementary Table 1). The results show no significant accuracy difference compared to LoRA, confirming that **orthogonal design in LoRA does not boost single-task performance**.
>
> We believe the **orthogonality design in LoRA excels in "multi-task" scenarios**—such as model merging (this work) and continual learning (O-LoRA [6])—because it **decouples parameter interference across multiple downstream tasks** when fine-tuning from the base model.
>
> **Supplementary Table 1: Single-task accuracy comparison (Pass@1 for HumanEval) using Qwen-2.5-7B.**
>
> |Method|MMLU|ScienceQA|GSM8K|HumanEval|
> |:-:|:-:|:-:|:-:|:-:|
> |LoRA (r=8)|49.84±0.56|92.84±0.13|77.01±0.32|47.20±1.54|
> |Ortho-LoRA (r=8)|49.74±0.73|92.82±0.28|77.06±0.19|47.06±0.98|
>
> **Typos:**
>
> Thanks for pointing it out. We will fix it in our the final version.
>
> **References**
>
> [1] Controlling Text-to-Image Diffusion by Orthogonal Finetuning. Neurips2023.
>
> [2] Parameter-Efficient Orthogonal Finetuning via Butterfly Factorization. ICLR2024.
>
> [3] ReFT: Representation Finetuning for Language Models. Neurips2024.
>
> [4] LoRA-FA: Memory-efficient Low-rank Adaptation for Large Language Models Fine-tuning. Arxiv2023.
>
> [5] Asymmetry in Low-Rank Adapters of Foundation Models. ICML2024.
>
> [6] Orthogonal Subspace Learning for Language Model Continual Learning. EMNLP findings2023.

---

### Note · Authors · 2025-08-12

We appreciate the reviewers' professional and insightful feedback, which has helped us improve our work. Here, we would like to summarize the additional work conducted during the discussion phase and highlight the primary contributions of our work.

1. In response to the questions from reviewers oMGp and ruPf, we performed additional experiments, including tests on initialization sensitivity, more comprehensive ablation studies, larger backbone models, more challenging datasets, comparisons with additional baselines, and a deeper analysis of the performance drop after merging. For reviewer vPRz's question, we provided further theoretical assumptions and justifications regarding why the implicit router works and how FlyLoRA aids merging. We also addressed the open problem raised by reviewer gKhQ concerning the orthogonality design in PEFT. We are pleased that our responses have addressed the reviewers' concerns.

2. Our work began with the observation in MoE-LoRA that, under the same rank budget, finer allocation of experts' rank leads to better performance through intra-task decoupling, but it is less efficient due to the increased parameter count for the router. Inspired by the fly olfactory circuit, we assumed that inner activation is already sufficient for routing. Thus, we unified the router and matrix $A$ (with $A$ also being random) into a single matrix, while keeping matrix $B$ rank-wise activated according to the inner activation intensity. Moreover, the random matrices $A_i$ and $A_j$ naturally possess orthogonality, which facilitates model merging between different components due to inter-task decoupling.

---

### Decision · Program_Chairs · 2025-09-17

**Decision:**

Accept (poster)

**Comment:**

In their paper, the authors introduce FlyLoRA, a new version of LoRA based on an implicit Mixture of Experts (MoE) model. FlyLoRA improves on the original method by: (i) Using an implicit routing system to activate experts for each rank in the up-projection matrix; (ii) Replacing the standard dense, updatable down-projection matrix with a fixed, sparse random projection.

All the reviewers are positive about the contributions of the papers, including: (1) the novelty of the proposed FlyLoRA; (2) the experiments are sufficient to support the claims about the interpretability and performance of the FlyLoRA; (3) the paper is written and presented clearly. After the rebuttal, most of the concerns of the reviewers were addressed, and all the reviewers are happy with the current stage of the paper.

In my opinion, the contributions and originality of the proposed FlyLoRA are sufficient for acceptance at NeurIPS. Therefore, I recommend accepting it in its current form. I encourage the authors to address the reviewers’ suggestions and integrate their feedback into the camera-ready version of their paper.